# WaMo: Wavelet-Enhanced Multi-Frequency Trajectory Analysis for Fine-Grained Text-Motion Retrieval

## Abstract

Text-Motion Retrieval (TMR) aims to retrieve 3D motion sequences semantically relevant to text descriptions. However, matching 3D motions with text remains highly challenging, primarily due to the intricate structure of human body and its spatial-temporal dynamics. Existing approaches often overlook these complexities, relying on general encoding methods that fail to distinguish different body parts and their dynamics, limiting precise semantic alignment. To address this, we propose WaMo, a new wavelet-based multi-frequency feature extraction framework. It fully captures joint-specific and time-varying motion details across multiple resolutions on body joints, extracting discriminative motion features to achieve fine-grained alignment with texts. WaMo has three key components: (1) Trajectory Wavelet Decomposition decomposes motion signals into frequency components that preserve both local kinematic details and global motion semantics. (2) Trajectory Wavelet Reconstruction uses learnable inverse wavelet transforms to reconstruct original joint trajectories from extracted features, ensuring the preservation of essential spatial-temporal information. (3) Disordered Motion Sequence Prediction reorders shuffled motion sequences to improve learning of inherent temporal coherence, enhancing motion-text alignment. Extensive experiments demonstrate WaMo's superiority, achieving 17.0% and 18.2% improvements in $Rsum$ on HumanML3D and KIT-ML datasets, respectively, outperforming existing state-of-the-art (SOTA) methods. Code will be open-sourced upon acceptance.

## 1 Introduction

Text-Motion Retrieval (TMR) (Petrovich et al., 2023; Yin et al., 2024; Yu et al., 2024) is an emerging cross-modal retrieval task that has drawn significant attention in recent years. The primary objective is to retrieve semantically relevant 3D motion sequences from a database based on user-provided text queries. However, the inherent complexity of human body structure (e.g., limbs, torso) and their distinct spatial-temporal dynamics makes the robust semantic alignment of 3D motion and text significantly more challenging than conventional cross-modal retrieval tasks (Gorti et al., 2022; Wu et al., 2023; Reddy et al., 2025). Thus, successfully addressing TMR depends on effectively modeling the intricate structure and temporal complexities of the motion modality.

Existing methods often overlook the inherent complexity of 3D motion sequences, relying on conventional coarse-grained methods for representation extraction. Many works (Petrovich et al., 2023; Yang et al., 2024b; Yu et al., 2024; Yin et al., 2024; Li et al., 2025) indiscriminately process the human motion in different parts and moments. They simply regard all joints as a whole and do not consider the variations between moments. While MoPa (Yu et al., 2024) uses patch-based motion modeling to capture local spatiotemporal structures, it pools joints into different parts to simplify the skeletons, limiting its effectiveness on dense skeletons. Such methods fail to adequately capture the fine-grained, joint-specific dynamics within motions, limiting precise semantic alignment with text descriptions. As illustrated in Figure 1, multi-scale wavelet decomposition of human joint signals displays substantially richer and more discriminative information. It demonstrates the necessity of refined motion signal processing to establish comprehensive correspondence with textual descriptions.

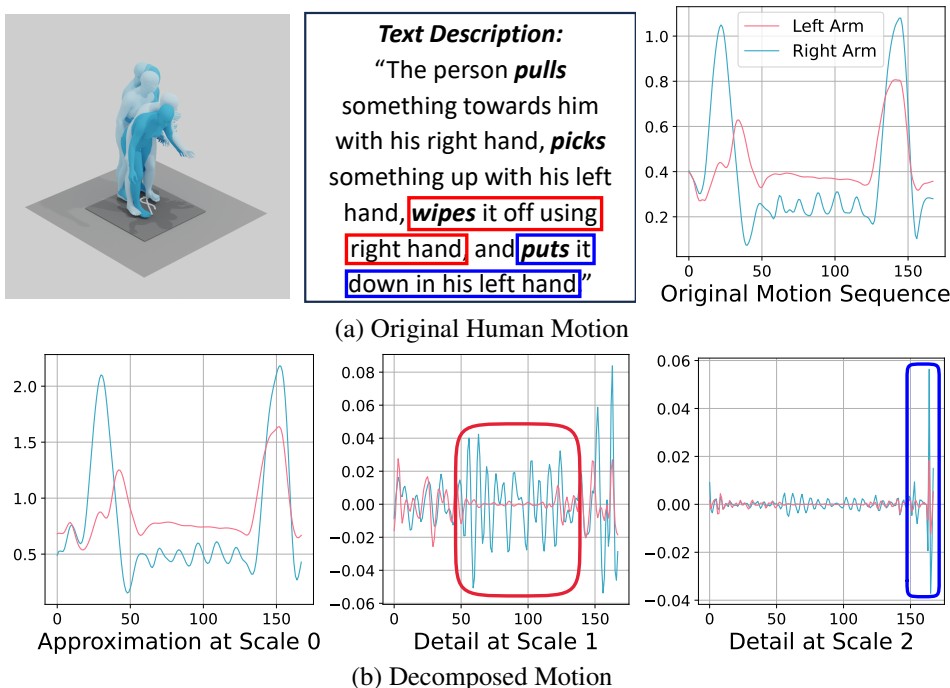

(a) Original Human Motion

(b) Decomposed Motion

Figure 1: **Wavelet decomposition of human motion signals.** For the original human motion trajectory of left and right arms (shown in the top-right subfigure, where the x-axis denotes time and y-axis denotes the amplitude of movements), we perform wavelet decomposition across three distinct scales. The scale-0 waveform preserves the overall structure of the original motion sequence. Compared to the scale-0 waveform, scale 1 highlights the rapid "wipe it off using right hand" movement (as in red boxes ), while scale 2 captures higher-frequency details of two-handed interactions related to "put it down in his left hand" (as in blue boxes ). It demonstrates how frequency-specific decomposition reveals the motion-text correspondence.

To address these limitations, we propose a comprehensive approach for processing kinematic information across multiple human joints through multi-frequency feature extraction. In particular, we adopt wavelet decomposition. Alternative methods such as convolutions primarily emphasize high frequencies due to their constrained receptive field (Wang et al., 2020). Prior works on single-modal motion prediction and generation have used Fourier Transform (FT) (Starke et al., 2020; 2022; Wan et al., 2023) or the Discrete Cosine Transform (DCT) (Mao et al., 2019; Chen et al., 2023) for human motion modeling. However, Fourier Transform utilizes a global frequency representation, thereby struggling to capture essential local temporal dynamics. DCT discards high-frequency components, which represent short and abrupt motions that are important in TMR. In contrast, wavelet decomposition effectively captures low-frequency components (Fujieda et al., 2018) while simultaneously preserving localized information (Finder et al., 2024). Technically, we extract low-frequency components to capture long-term movement trends, while simultaneously obtaining high-frequency features to represent short and abrupt motions. This multi-frequency analysis results in discriminative motion features that facilitate fine-grained alignment with textual semantics. Unlike prior methods (shengchuan gao et al., 2025; Feng et al., 2024) that do not distinguish the fine-grained semantics of low- and high-frequency components, we aim to capture both frequency-specific characteristics and their inter-dependencies, facilitating fine-grained alignment with textual semantics. To further improve the feature extraction process, we propose a learnable inverse wavelet transform module, which reconstructs the original joint trajectories from motion features. It ensures that the extractor has effectively captured motion characteristics inherent in the original joint movements.

Moreover, since text-described actions naturally follow a temporal order aligned with motion sequences, the model needs to capture the temporal dynamics of 3D motions. To enhance the understanding of temporal dynamics, we introduce a sequence reordering task. It involves reconstructing the correct order from randomly shuffled motion sequences, which encourages the model to learn

and maintain the temporal dependencies in the motions. As a result, it improves the model's ability to align motion with textual descriptions.

Concretely, we introduce WaMo, a novel wavelet-based multi-frequency feature extraction framework for TMR. Our framework consists of three key components: (1) Trajectory Wavelet Decomposition (TWD) decomposes motion signals into frequency components that preserve both local kinematic details and global motion semantics. By designing a special Intra- and Inter-Frequency Attention module, it can better capture both frequency-specific characteristics and their inter-dependencies. (2) Trajectory Wavelet Reconstruction (TWR) introduces a constraint using learnable inverse wavelet transforms to reconstruct the original joint trajectory from the extracted features. It acts as a regularization term to make sure the motion encoder extracts more pertinent kinematical information. (3) Disordered Motion Sequence Prediction (DMSP) randomly shuffles motion sequences and trains the model to recover their original temporal orders. This strategy explicitly enforces the learning of inherent motion dynamics, improving the alignment between sequential motion dynamics and textual descriptions.

In summary, our key contributions are as follows:

- We introduce WaMo, a novel wavelet-based multi-frequency feature extraction framework. It captures both intra-frequency characteristics and inter-frequency dependencies across multiple scales via wavelet transforms, leading to refined 3D motion representations.
- We propose the trajectory wavelet reconstruction module, using learnable inverse wavelet transforms to reconstruct original human motion trajectories from extracted features. It ensures the preservation of spatial-temporal information within features and enhances the feature robustness and discrimination.
- Extensive experiments on two datasets, HumanML3D (Guo et al., 2022) and KIT-ML (Plappert et al., 2016), validate the superiority of our method. WaMo outperforms existing state-of-the-art methods by substantial margins, achieving 17.0% and 18.2% improvements in $Rsum$ on these datasets, respectively.

## 2 RELATED WORKS

### 2.1 TEXT-MOTION RETRIEVAL

In recent years, text-motion retrieval has received much attention, which aims to achieve mutual matching between text descriptions and 3D human motions. In particular, TMR (Petrovich et al., 2023) extends the text-to-motion generation model TEMOS (Petrovich et al., 2022) with a contrastive loss. MoPa (Yu et al., 2024) regards XYZ coordinates as RGB pixels to encode 3D motions as 2D images. LAVIMO (Yin et al., 2024) introduces videos as an additional modality to bridge the gap between texts and motions. MGSI (Yang et al., 2024b) formulates text-motion retrieval as a multi-instance multi-label learning problem. MESM (Shi & Zhang, 2024) adopts a large language model to expand coarse text descriptions into fine-grained ones. LaMP (Li et al., 2025) conducts joint training for text-motion retrieval, text-to-motion generation, and motion-to-text captioning. However, they adopt coarse-grained methods to extract motion representations, overlooking the intricate nature of human joints. In contrast, we adopt multi-frequency wavelet transforms to decompose human motions, leading to more detailed information.

### 2.2 CROSS-MODAL SEMANTIC ALIGNMENT

Cross-modal semantic alignment is a critical yet challenging task that aims to align different modalities in the latent semantic space (Zhou et al., 2021; Wu et al., 2022; Jin et al., 2023; Fu et al., 2024; Feng et al., 2025). DE++ (Dong et al., 2021) learns global and local patterns in latent space and concept space. VT-TWINS (Ko et al., 2022) aligns noisy and weakly correlated multi-modal time-series data using differentiable Dynamic Time Warping.TRM (Zheng et al., 2023) mines phrase-level temporal relationships between videos and sentences. SSN (Yang et al., 2024a) decomposes semantic shifts within modalities to guide cross-modal alignment. CMA (Kim & Hwang, 2025) regularizes the distances between image and text embeddings in the hyper-spherical representation space. Generally, existing methods mainly focus on 2D images or videos. Different from them, we perform cross-modal alignment on complex 3D motions and texts.

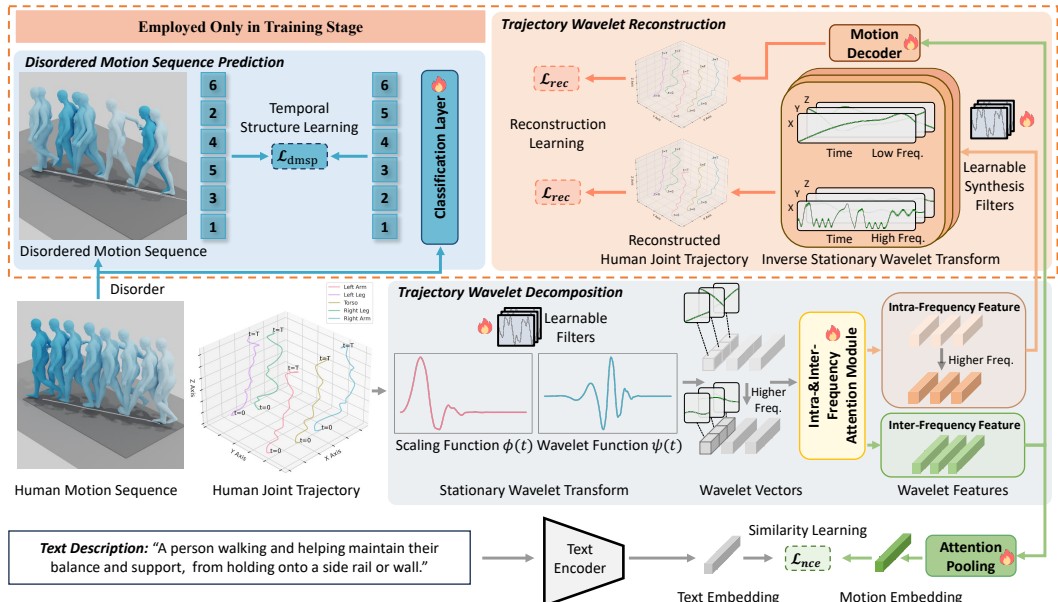

Figure 2: **The overview of WaMo.** We comprehensively model the intricate spatial-temporal dynamics of 3D human motion and establish precise alignment with textual semantics. WaMo consists of three key components: (1) **Trajectory Wavelet Decomposition** (TWD) decomposes motion signals into frequency components that preserve both local kinematic details and global motion semantics. It enables comprehensive multi-scale frequency analysis. (2) **Trajectory Wavelet Reconstruction** (TWR) facilitates accurate reconstruction of original motion trajectories, thereby ensuring robust motion representation learning. (3) **Disordered Motion Sequence Prediction** (DMSP) enhances temporal understanding by training the model to recover correct temporal ordering from shuffled motion sequences, improving the modeling of motion dynamics.

## 3 PRELIMINARIES

Wavelet decomposition is widely adopted to analyze signals at multiple levels of detail (Mallat, 2002; Guo et al., 2017; Yao et al., 2022; Oka et al., 2025). Unlike Fast Fourier Transform (FFT), which primarily focuses on global frequency representation, wavelet decomposition captures both global and local temporal dynamics. In particular, Discrete Wavelet Transform (DWT) conducts shift-variant decomposition, disrupting the original temporal structure. Therefore, we employ Stationary Wavelet Transform (SWT), which captures multi-scale temporal dynamics while preserving the original temporal structure of signals.

### 3.1 STATIONARY WAVELET TRANSFORM

Let $x[n]$ be a discrete time signal and $h[k]$, $g[k]$ the low–pass and high–pass filters obtained by scaling function $\phi(t)$ and wavelet function $\psi(t)$, where $k$ specifies the translation. At level $s$, approximation coefficients $a_s$ and detail coefficients $d_s$ are computed as:

$$a_s[n] = \sum_k h[k]\, a_{s-1}[n + 2^{s-1}k], \; d_s[n] = \sum_k g[k]\, a_{s-1}[n + 2^{s-1}k], \qquad (1)$$

with the initialization $a_0[n] = x[n]$. The complete SWT decomposition of level $S$ recursively produces the coefficient set $\{d_1, d_2, \ldots, d_S, a_S\}$.

### 3.2 INVERSE STATIONARY WAVELET TRANSFORM

Let $\tilde{h}[k]$, $\tilde{g}[k]$ be the synthesis filters forming a precise-reconstruction pair with $h, g$. Starting from $a_S$ and the detail sequences $d_s$, the reconstructed low-pass signal at level $s - 1$ is:

$$a_{s-1}[n] = \sum_k \tilde{h}[k]\, a_s[n - 2^{s-1}k] + \sum_k \tilde{g}[k]\, d_s[n - 2^{s-1}k], \qquad (2)$$

After $S$ levels, the original signal is recovered: $\hat{x}[n] = a_0[n]$.

## 4 METHOD

### 4.1 OVERVIEW

The overall framework of WaMo is illustrated in Figure 2. Our method employs wavelet transforms to comprehensively analyze the spatial-temporal characteristics of human joint movements and establish robust correspondence with textual semantics. In Section 4.2, we first detail our text encoder. We then present the Trajectory Wavelet Decomposition (TWD) to encode motion sequence signals in Section 4.3, which decomposes motion signals into multi-scale frequency components to fully capture motion dynamics. The Trajectory Wavelet Reconstruction (TWR) is subsequently introduced in Section 4.4, ensuring the preservation of spatial-temporal information. Finally, we present Disordered Motion Sequence Prediction (DMSP) in Section 4.5, where the model learns temporal structures by recovering the correct ordering of shuffled motion sequences.

### 4.2 TEXT ENCODER

For the textual representation, we adopt the pre-trained DistilBERT (Sanh et al., 2019) to encode text embeddings, following prior works (Petrovich et al., 2023; Yu et al., 2024; Shi & Zhang, 2024). The output of the [CLS] token is projected into the latent semantic space of motion-language modalities. The encoded text embedding is denoted as $t \in \mathbb{R}^D$, where $D$ is the latent dimension.

### 4.3 TRAJECTORY WAVELET DECOMPOSITION

Given a motion sequence with $T$ frames and $J$ joints in $xyz$ coordinates, we denote it as $M \in \mathbb{R}^{T \times J \times 3}$. To decompose it into frequency components that preserve both local kinematic details and global motion semantics, we adopt learnable SWT (Nason & Silverman, 1995). It provides a time-invariant decomposition and preserves the original temporal structure. SWT is performed along time per joint coordinate, leading to low-frequency and high-frequency vectors $M_{low} \in \mathbb{R}^{T \times J \times 3}$ and $M_{high} \in \mathbb{R}^{S \times T \times J \times 3}$:

$$M \xrightarrow{SWT(h_0, g_0)} \{M_{\text{low}}, M_{\text{high}}\}, \tag{3}$$

where $h_0$ and $g_0$ are learnable low-pass and high-pass filters, respectively, and $S$ is the decomposition level. $M_{low}$ and $M_{high}$ are extracted with temporal patterns at distinct scales, capturing both local/global patterns across multiple levels.

#### 4.3.1 INTRA- AND INTER-FREQUENCY ATTENTION

To fully capture comprehensive motion characteristics across diverse temporal scales, we propose the Intra- and Inter-Frequency Attention module. The pipeline is shown in Figure 3. For each frequency component, we first employ specialized 1D convolutions: long-term temporal patterns in $M_{low}$ are captured using large kernels, while short-term variations in $M_{high}$ are extracted via small kernels. The processed features are then flattened and mapped by a multi-layered perceptron (MLP), followed

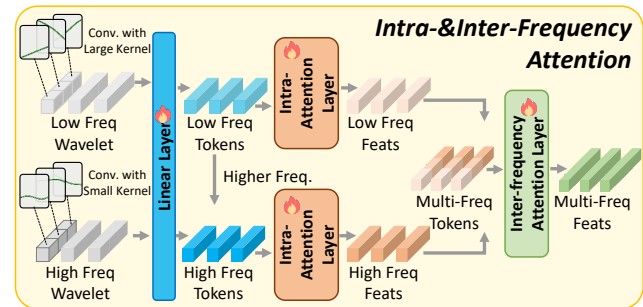

Figure 3: The pipeline of the Intra- and Inter-Frequency Attention module. It preserves frequency-specific characteristics while capturing their inter-dependencies.

by a Transformer encoder (Vaswani et al., 2017). This yields the intra-frequency representations $\hat{M}_{low} \in \mathbb{R}^{T \times D}$ for global motion trends and $\hat{M}_{high} \in \mathbb{R}^{S \times T \times D}$ for local kinematic details.

To integrate multi-scale motion characteristics, we concatenate the intra-frequency features along the feature dimension, forming the combined representation $\hat{M}_{multi} \in \mathbb{R}^{T \times (S+1)D}$. This fused

feature captures both global motion trends ($\hat{M}_{low}$) and local kinematic details ($\hat{M}_{high}$). It is then processed through an MLP and Transformer layer to compute the final inter-frequency feature $\hat{M} \in \mathbb{R}^{T \times D}$, which comprehensively encodes the multi-resolution spatial-temporal dynamics of human motion. Finally, we employ the additive attention pooling (Bahdanau et al., 2015) on $\hat{M}$ to obtain the aggregated motion embedding $m \in \mathbb{R}^D$. To align the text and motion embeddings, following prior works (Petrovich et al., 2023; Yu et al., 2024), we adopt the InfoNCE loss (Oord et al., 2018):

$$\mathcal{L}_{nce} = -\frac{1}{B} \sum_i \left( \log \frac{\exp(S_{ii}/\tau)}{\sum_j \exp(S_{ij}/\tau)} + \log \frac{\exp(S_{ii}/\tau)}{\sum_j \exp(S_{ji}/\tau)} \right), \tag{4}$$

where $B$ is the size of the mini-batch, $S_{ij} = cos(t_i, m_j)$ is the cosine similarity of the $i$-th text and $j$-th motion embeddings within the mini-batch, and $\tau$ is the temperature hyperparameter.

### 4.4 TRAJECTORY WAVELET RECONSTRUCTION

To ensure the encoded intra- and inter-frequency features preserve the original spatial-temporal structures of motions, we further introduce the Trajectory Wavelet Reconstruction (TWR) module. The intra-frequency features $\hat{M}_{low}$ and $\hat{M}_{high}$ are transformed through separate MLPs to recover low- and high-frequency vectors $\bar{M}_{low} \in \mathbb{R}^{T \times J \times 3}$ and $\bar{M}_{high} \in \mathbb{R}^{S \times T \times J \times 3}$. Then, we employ inverse SWT (ISWT) to reconstruct the motion sequence, which is denoted as $\bar{M}_{intra} \in \mathbb{R}^{T \times J \times 3}$:

$$\left\{ \bar{M}_{low}, \bar{M}_{high} \right\} \xrightarrow{ISWT(h_1, g_1)} \bar{M}_{intra}, \tag{5}$$

where $h_1$ and $g_1$ are learnable low-pass and high-pass synthesis filters. For the inter-frequency feature $\hat{M}$, we directly input it into an MLP motion decoder to obtain the holistic reconstructed motion $\bar{M}_{inter} \in \mathbb{R}^{T \times J \times 3}$, as it already incorporates cross-frequency correlations. We compare these reconstructed motions to the ground-truth motion $M$ via:

$$\mathcal{L}_{rec} = \mathcal{L}_1(\bar{M}_{intra}, M) + \mathcal{L}_1(\bar{M}_{inter}, M), \tag{6}$$

where $\mathcal{L}_1$ is the smooth L1 loss (Girshick, 2015).

### 4.5 DISORDERED MOTION SEQUENCE PREDICTION

Inspired by self-supervised Jigsaw Puzzle Solving in image representation learning (Wei et al., 2019; Jing & Tian, 2020; Zhuge et al., 2021), we propose the Disordered Motion Sequence Prediction (DMSP) module to learn temporal structures inherent in motion sequences. However, given the fundamental differences between 2D image/video and 3D human motion, directly applying 2D sequence shuffling techniques (Lee et al., 2017) could indiscriminately rearrange frames across the entire sequence. Such methods could disrupt local kinematic dependencies, resulting in discontinuous and physically impossible actions that confuse the encoder. In contrast, DMSP is specifically designed for the unique dynamics of motion. By dividing sequences into temporally coherent groups and only shuffling a small group of frames, DMSP preserves the local kinematic continuity while forcing the model to reconstruct the global semantic causal order. We provide experiments comparing 2D sequence shuffling techniques (Lee et al., 2017) and our DMSP in Section 5.4.

Technically, we first divide the motion sequence into $\lambda_g$ temporally coherent groups, where each group contains $T/\lambda_g$ consecutive frames sharing the same temporal label. Then, we randomly select and shuffle a fraction $\lambda_s$ of frames to create the shuffled sequence, followed by the TWD module to obtain the disrupted motion feature $\bar{M} \in \mathbb{R}^{T \times D}$. The model then learns temporal structures by predicting the original and shuffled group labels $g_o, g_s \in \mathbb{R}^T$:

$$p_o = Softmax(CLS(\hat{M})), \quad p_s = Softmax(CLS(\bar{M})), \tag{7}$$

where $CLS$ is the classification layer, $p_o \in \mathbb{R}^{T \times \lambda_g}$ and $p_s \in \mathbb{R}^{T \times \lambda_g}$ are the predicted probability distributions of group labels. The training objective is defined as:

$$\mathcal{L}_{dmsp} = \mathcal{L}_{CE}(p_o, g_o) + \mathcal{L}_{CE}(p_s, g_s), \tag{8}$$

where $\mathcal{L}_{CE}$ represents the cross-entropy loss function.

| Method | Venue | Text-to-Motion | | | | | | Motion-to-Text | | | | | | Rsum↑ |
|--------|-------|------|------|------|------|-------|-------|------|------|------|------|-------|-------|-------|
| | | R@1↑ | R@2↑ | R@3↑ | R@5↑ | R@10↑ | MedR↓ | R@1↑ | R@2↑ | R@3↑ | R@5↑ | R@10↑ | MedR↓ | |
| *Text-Motion Generation Models:* | | | | | | | | | | | | | | |
| T2M | CVPR'22 | 1.80 | 3.42 | 4.79 | 7.12 | 12.47 | 81.00 | 2.92 | 3.74 | 6.00 | 8.36 | 12.95 | 81.50 | 63.57 |
| TEMOS | ECCV'22 | 2.12 | 4.09 | 5.87 | 8.26 | 13.52 | 173.00 | 3.86 | 4.54 | 6.94 | 9.38 | 14.00 | 183.25 | 72.58 |
| MotionCLIP | ECCV'22 | 2.33 | 5.85 | 8.93 | 12.77 | 18.14 | 103.00 | 5.12 | 6.97 | 8.35 | 12.46 | 19.02 | 91.42 | 99.94 |
| MoMask† | CVPR'24 | 2.00 | 3.58 | 5.03 | 7.42 | 13.17 | 77.00 | 1.81 | 3.58 | 5.22 | 7.87 | 13.31 | 77.00 | 62.99 |
| *Text-Motion Retrieval Models:* | | | | | | | | | | | | | | |
| MoT | SIGIR'23 | 2.61 | 4.72 | 6.90 | 10.66 | 17.79 | 60.00 | 4.03 | 5.07 | 7.43 | 11.23 | 17.68 | 64.25 | 88.12 |
| TMR | ICCV'23 | 5.68 | 10.59 | 14.04 | 20.34 | 30.94 | 28.00 | 9.95 | 12.44 | 17.95 | 23.56 | 32.69 | 28.50 | 178.18 |
| MGSI | MM'24 | 6.61 | 12.73 | 17.11 | 23.91 | 34.74 | 24.00 | 10.61 | 13.18 | 19.75 | 26.00 | 36.63 | 22.50 | 201.27 |
| MESM | MM'24 | 7.16 | 12.52 | 16.70 | 24.22 | 35.38 | 23.00 | 11.19 | 13.81 | 19.59 | 25.96 | 35.93 | 23.25 | 202.46 |
| LAVIMO | CVPR'24 | 6.37 | 11.84 | 15.60 | 21.95 | 33.67 | 24.00 | 9.72 | 13.33 | 18.73 | 25.00 | 36.55 | 22.50 | 192.76 |
| MoPa | CVPR'24 | 10.80 | 14.98 | 20.00 | 26.72 | 38.02 | 19.00 | 11.25 | 13.86 | 19.98 | 26.86 | 37.40 | 20.50 | 219.87 |
| LaMP† | ICLR'25 | 3.65 | 7.56 | 10.90 | 16.56 | 26.26 | 30.00 | 4.51 | 8.71 | 11.76 | 17.34 | 27.84 | 30.00 | 135.09 |
| **WaMo (Ours)** | - | **14.02** | **17.58** | **25.51** | **32.06** | **42.10** | **16.00** | **15.51** | **16.57** | **22.74** | **29.40** | **41.73** | **17.00** | **257.22** |

Table 1: **Comparison results on HumanML3D (Guo et al., 2022).** The best results are shown in **bold** and the second-best outcomes are underlined. We achieve state-of-the-art results across all metrics. † denotes reproduced using official checkpoints.

| Method | Venue | Text-to-Motion | | | | | | Motion-to-Text | | | | | | Rsum↑ |
|--------|-------|------|------|------|------|-------|-------|------|------|------|------|-------|-------|-------|
| | | R@1↑ | R@2↑ | R@3↑ | R@5↑ | R@10↑ | MedR↓ | R@1↑ | R@2↑ | R@3↑ | R@5↑ | R@10↑ | MedR↓ | |
| *Text-Motion Generation Models:* | | | | | | | | | | | | | | |
| T2M | CVPR'22 | 3.37 | 6.99 | 10.84 | 16.87 | 27.71 | 28.00 | 4.94 | 6.51 | 10.72 | 16.14 | 25.30 | 28.50 | 129.39 |
| TEMOS | ECCV'22 | 7.11 | 13.25 | 17.59 | 24.10 | 35.66 | 24.00 | 11.69 | 15.30 | 20.12 | 26.63 | 36.39 | 26.50 | 207.84 |
| MotionCLIP | ECCV'22 | 4.87 | 9.31 | 14.36 | 20.09 | 31.57 | 26.00 | 6.55 | 11.28 | 17.12 | 25.48 | 34.97 | 23.00 | 175.60 |
| MoMask† | CVPR'24 | 3.69 | 7.81 | 12.22 | 17.47 | 29.83 | 22.00 | 4.55 | 8.10 | 11.22 | 17.19 | 28.69 | 22.00 | 140.77 |
| *Text-Motion Retrieval Models:* | | | | | | | | | | | | | | |
| MoT | SIGIR'23 | 6.23 | 11.07 | 16.54 | 23.92 | 37.15 | 20.00 | 10.56 | 13.49 | 20.61 | 27.61 | 38.04 | 19.50 | 205.22 |
| TMR | ICCV'23 | 7.23 | 13.98 | 20.36 | 28.31 | 40.12 | 17.00 | 11.20 | 13.86 | 20.12 | 28.07 | 38.55 | 18.00 | 221.80 |
| MGSI | MM'24 | 8.91 | 16.28 | 20.87 | 29.64 | 40.84 | 16.00 | 13.49 | 16.41 | 23.54 | 30.66 | 43.00 | 15.50 | 243.64 |
| MESM | MM'24 | 9.29 | 17.05 | 22.31 | 29.13 | 41.02 | 16.00 | 12.75 | 16.41 | 24.17 | 32.59 | 42.88 | 15.50 | 247.60 |
| LAVIMO | CVPR'24 | 10.16 | 19.92 | 24.61 | 34.57 | 49.80 | 11.00 | 15.43 | 20.12 | 26.95 | 34.57 | 53.32 | 10.00 | 289.45 |
| MoPa | CVPR'24 | 14.02 | 21.08 | 28.91 | 34.10 | 50.00 | 10.50 | 13.61 | 17.26 | 27.54 | 33.33 | 44.77 | 13.00 | 284.62 |
| LaMP† | ICLR'25 | 6.25 | 12.50 | 17.76 | 24.86 | 42.47 | 13.00 | 6.39 | 14.49 | 18.89 | 27.56 | 44.03 | 13.00 | 215.20 |
| **WaMo (Ours)** | - | **18.31** | **24.82** | **34.46** | **43.49** | **56.75** | **8.00** | **19.04** | **22.41** | **31.11** | **37.83** | **54.04** | **9.00** | **342.26** |

Table 2: **Comparison results on KIT-ML (Plappert et al., 2016).** The best results are shown in **bold** and the second-best outcomes are underlined. We achieve state-of-the-art results across all metrics. † denotes reproduced using official checkpoints.

## 5 EXPERIMENTS

### 5.1 EXPERIMENT SETUP

#### 5.1.1 DATASETS

We employ two widely-adopted datasets to validate the proposed method: HumanML3D (Guo et al., 2022) and KIT-ML (Plappert et al., 2016). **HumanML3D** contains human motions from AMASS (Mahmood et al., 2019) and HumanAct12 (Guo et al., 2020) with additional text descriptions. Following the official split, the training, validation, and test sets have 23384, 1460, and 4380 motions, respectively. Each motion is annotated with 3 text captions on average. **KIT-ML** primarily focuses on locomotion. It is split into 4888, 300, and 830 motions for training, validation, and test sets, respectively. Each motion is associated with an average of 2.1 text captions.

#### 5.1.2 METRICS

We adopt the commonly used Recall at k (R@K), Median Rank (MedR), and Rsum as metrics. R@K is the ratio of queries that retrieve target items in the top-K results. K is set to 1,2,3,5,10. MedR is the median rank of target items. Rsum is the sum of R@K values. Higher R@K, Rsum, and lower MedR indicate better retrieval accuracy.

### 5.1.3 IMPLEMENTATION DETAILS

In the experiments, we adopt the Adam optimizer (Kingma & Ba, 2015) with a learning rate of 1e-4 and a cosine annealing schedule to train the model. The latent dimension $D$, decomposition level $S$, and temperature $\tau$ are set to 256, 3, and 0.07, respectively. The group number $\lambda_g$ and shuffle ratio $\lambda_s$ are set to 16 and 0.25. For fair comparison with prior works, we follow the same motion pre-processing method proposed in T2M (Guo et al., 2022). Unless otherwise specified, we mainly report text-to-motion retrieval results.

## 5.2 COMPARISON WITH STATE-OF-THE-ARTS

We compare our method with the following state-of-the-arts (SOTA): text-motion generation methods, *i.e.,* T2M (Guo et al., 2022), TEMOS (Petrovich et al., 2022), MotionCLIP (Tevet et al., 2022), MoMask (Guo et al., 2024), and text-motion retrieval methods, *i.e.,* MoT (Messina et al., 2023), TMR (Petrovich et al., 2023), MGSI (Yang et al., 2024b), MESM (Shi & Zhang, 2024), LAV-IMO (Yin et al., 2024), MoPa (Yu et al., 2024), and LaMP (Li et al., 2025). As shown in Table 1 and 2, our method outperforms prior works by a substantial margin across all metrics on both datasets. Notably, compared to prior SOTA methods, MoPa and LAVIMO, we achieve 17.0% and 18.2% relative improvements on the Rsum metric of HumanML3D and KIT datasets, respectively. These results demonstrate the superior effectiveness of our approach.

## 5.3 ABLATION STUDIES

### 5.3.1 MAIN ABLATION STUDIES

In Table 3, we comprehensively assess the effectiveness of the proposed modules. Concretely, when TWD is removed, we directly flatten and map all human joint coordinates in one frame into a token, followed by a Transformer encoder to encode all frames (tokens). To ablate TWR or DMSP, we just remove $\mathcal{L}_{rec}$ or $\mathcal{L}_{dmsp}$, respectively. From the table, it is found that the introduction of each component can bring performance improvements, and the best performance is achieved when all modules are introduced (Row 6).

| Row | Setting | | | HumanML3D | | | KIT-ML | | |
|-----|---------|---------|------|-------|-------|-------|-------|-------|-------|
| | *TWD* | *TWR* | *DMSP* | R@1↑ | R@2↑ | R@3↑ | R@1↑ | R@2↑ | R@3↑ |
| 1 | ✗ | ✗ | ✗ | 9.14 | 13.65 | 18.72 | 12.77 | 21.08 | 25.30 |
| 2 | ✓ | ✗ | ✗ | 11.91 | 14.27 | 20.68 | 15.78 | 21.70 | 29.89 |
| 3 | ✗ | ✗ | ✓ | 11.36 | 15.18 | 21.59 | 15.23 | 22.53 | 28.92 |
| 4 | ✓ | ✗ | ✓ | 12.43 | 15.85 | 22.17 | 16.43 | 22.70 | 31.93 |
| 5 | ✓ | ✓ | ✗ | 12.24 | 15.87 | 22.46 | 16.99 | 23.13 | 31.08 |
| 6 | ✓ | ✓ | ✓ | **14.02** | **17.58** | **25.51** | **18.31** | **24.82** | **34.46** |

Table 3: **Main ablation studies of the proposed key modules on text-to-motion retrieval.** Note that without TWD, TWR is also disabled, since the inverse wavelet transform process is based on the wavelet decomposition process.

### 5.3.2 FREQUENCY DOMAIN TRANSFORM METHOD

In Table 4, we compare the performance of different frequency domain transform methods, including learnable convolutions, FFT, DCT, DWT, and SWT. Correspondingly, we also adopt parameter-matched convolutions, inverse FFT, DCT, DWT, and SWT for reconstruction in TWR. For fair comparison, we maintain the same number of Transformer layers to encode motions. In the baseline method (Row 1), no transform is used. As shown in Table 4, SWT no-

| Row | Method | HumanML3D | | | KIT-ML | | |
|-----|--------|-------|-------|-------|-------|-------|-------|
| | | R@1↑ | R@2↑ | R@3↑ | R@1↑ | R@2↑ | R@3↑ |
| 1 | N.A. | 11.36 | 15.18 | 21.59 | 15.23 | 22.53 | 28.92 |
| 2 | FFT | 12.10 | 15.44 | 22.07 | 15.98 | 23.21 | 30.62 |
| 3 | DCT | 12.69 | 16.10 | 22.84 | 16.17 | 23.96 | 31.15 |
| 4 | Conv | 12.86 | 15.97 | 23.40 | 16.63 | 24.06 | 31.08 |
| 5 | DWT | 13.27 | 16.12 | 23.91 | 16.73 | 23.99 | 32.59 |
| 5 | SWT | **14.02** | **17.58** | **25.51** | **18.31** | **24.82** | **34.46** |

Table 4: **Ablation study on the frequency domain transform method.** *N.A.* denotes do not apply transforms.

tably outperforms all other settings across all metrics. Although surpassing the baseline, FFT performs worst due to its global frequency representation, which struggles to capture the essential local temporal dynamics. Convolutions and DCT lead to inferior results compared to DWT and SWT,

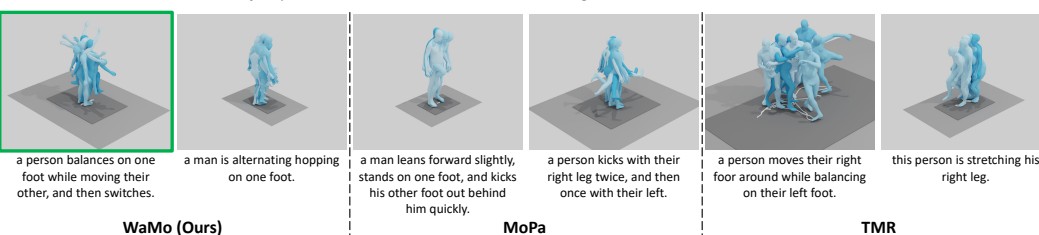

Query: a person **balances** on one foot while **moving** their other, and then **switches**.

| a person balances on one foot while moving their other, and then switches. | a man is alternating hopping on one foot. | a man leans forward slightly, stands on one foot, and kicks his other foot out behind him quickly. | a person kicks with their right leg twice, and then once with their left. | a person moves their right foor around while balancing on their left foot. | this person is stretching his right leg. |
|---|---|---|---|---|---|
| **WaMo (Ours)** | | **MoPa** | | **TMR** | |

Figure 5: **Visualization comparisons of fine-grained text-to-motion retrieval between our method, MoPa (Yu et al., 2024), and TMR (Petrovich et al., 2023) on HumanML3D.** Top-2 retrieval results are shown from left to right. The ground-truth motion is highlighted by a green box .

primarily due to the inadequate consideration of low-frequency and high-frequency components, respectively. DWT further shows moderate improvements but suffers from shift-variant decomposition, disrupting the original temporal structure. In contrast, SWT can capture multi-resolution motion semantics while preserving the temporal structure, thus leading to the best performance.

### 5.3.3 DECOMPOSITION LEVEL

In Table 5, we investigate the influence of the decomposition level $S$ on retrieval performance and reconstruction quality. It is found that the performance consistently increases from $S=1$ to $S=3$ across all retrieval metrics. FID also decreases significantly, indicating less information loss. It demonstrates that deeper decomposition captures richer temporal hierarchies. However, the performance slightly drops and FID increases when $S=4$, due to excessive decomposition may introduce noisy and motion-irrelevant features, interfering with the precise matching with texts and motion reconstruction.

| $S$ | HumanML3D | | | | KIT-ML | | | |
|---|---|---|---|---|---|---|---|---|
| | R@1↑ | R@2↑ | R@3↑ | FID↓ | R@1↑ | R@2↑ | R@3↑ | FID↓ |
| 1 | 11.93 | 15.75 | 22.21 | 0.049 | 15.66 | 22.85 | 30.26 | 0.152 |
| 2 | 12.81 | 16.39 | 23.16 | 0.032 | 16.99 | 23.73 | 30.96 | 0.127 |
| 3 | **14.02** | **17.58** | **25.51** | **0.018** | **18.31** | **24.82** | **34.46** | **0.085** |
| 4 | 13.84 | 17.20 | 24.52 | 0.026 | 18.19 | 24.06 | 31.33 | 0.112 |

Table 5: Ablations on decomposition level $S$.

### 5.3.4 HYPERPARAMETERS $\lambda_g$ AND $\lambda_s$

In Figure 4, we show the sensitivity of our model to group number $\lambda_g$ and shuffle ratio $\lambda_s$. Optimal results are achieved when $\lambda_g=16$ and $\lambda_s=0.25$ on both datasets. Particularly, the performances are stable and exhibit limited fluctuations across various $\lambda_g$ and $\lambda_s$ choices, indicating the robustness and insensitiveness of our method to different hyperparameters.

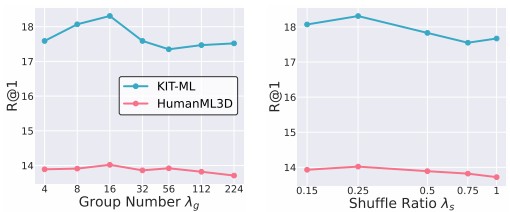

Figure 4: Ablation study on the group number $\lambda_g$ and shuffle ratio $\lambda_s$.

### 5.3.5 INTRA- AND INTER-FREQUENCY ATTENTION

In Table 6, we ablate the contribution of the Intra- or Inter-Frequency Attention. Note that we maintain the same number of Transformer layers in all experiments to ensure fair comparison. It is found that solely intra-frequency attention outperforms inter-frequency (Row 1 and 2) due to its capture of fine-grained motion semantics across diverse temporal scales. Moreover, their joint application (Row 3) yields further performance improvements since the cross-frequency fusion integrates complementary information.

| Row | Attention | | HumanML3D | | | KIT-ML | | |
|---|---|---|---|---|---|---|---|---|
| | intra | inter | R@1↑ | R@2↑ | R@3↑ | R@1↑ | R@2↑ | R@3↑ |
| 1 | ✓ | ✗ | 13.17 | 16.12 | 24.02 | 17.71 | 22.97 | 31.08 |
| 2 | ✗ | ✓ | 12.82 | 15.74 | 23.36 | 17.11 | 22.71 | 30.12 |
| 3 | ✓ | ✓ | **14.02** | **17.58** | **25.51** | **18.31** | **24.82** | **34.46** |

Table 6: Ablations on Intra- or Inter-Frequency Attention.

### 5.3.6 TRAJECTORY WAVELET RECONSTRUCTION

In Table 7, we demonstrate the effectiveness of Trajectory Wavelet Reconstruction. Both intra- and inter-frequency reconstruction result in improvements on all metrics (Row 1-3), since these reconstruction constraints enforce the motion encoder to extract more relevant frequency-localized details and holistic motion semantics corresponding to the text. Their combination further leads to the

| Row | Reconstruction | | HumanML3D | | | KIT-ML | | |
|-----|----------------|-------|-------|-------|-------|-------|-------|-------|
| | *intra* | *inter* | R@1↑ | R@2↑ | R@3↑ | R@1↑ | R@2↑ | R@3↑ |
| 1 | ✘ | ✘ | 12.43 | 15.85 | 22.17 | 16.43 | 22.70 | 31.93 |
| 2 | ✓ | ✘ | 13.27 | 16.42 | 23.94 | 17.47 | 23.46 | 33.18 |
| 3 | ✘ | ✓ | 13.11 | 16.21 | 23.89 | 17.22 | 23.14 | 32.97 |
| 4 | ✓ | ✓ | **14.02** | **17.58** | **25.51** | **18.31** | **24.82** | **34.46** |

Table 7: Ablations on Trajectory Wavelet Reconstruction.

best performance, indicating the complementary nature of both localized and holistic processing. It verifies the reasonability of proposed methods.

### 5.4 DISORDERED MOTION SEQUENCE PREDICTION

To quantitatively verify the effectiveness of the Disordered Motion Sequence Prediction (DMSP) module, we compare our method with two variants: one baseline without sequence shuffling, and the other one replacing DMSP with the 2D video shuffling technique (Lee et al., 2017). As shown in Table 8, the video shuffling method significantly underper-

| Setting | HumanML3D | | | KIT-ML | | |
|---------|-------|-------|-------|-------|-------|-------|
| | R@1↑ | R@2↑ | R@3↑ | R@1↑ | R@2↑ | R@3↑ |
| WaMo w/o shuffling | 12.24 | 15.87 | 22.46 | 16.99 | 23.13 | 31.08 |
| WaMo w/ video shuffling | 11.56 | 15.42 | 21.55 | 15.03 | 22.59 | 28.89 |
| WaMo w/ DMSP (Ours) | **14.02** | **17.58** | **25.51** | **18.31** | **24.82** | **34.46** |

Table 8: Ablation study on the Disordered Motion Sequence Prediction module.

forms our DMSP and even falls behind the baseline. It indicates that preserving local continuity is critical for learning motion representations.

### 5.5 QUALITATIVE RESULTS

We present a comparative visualization of fine-grained text-to-motion retrieval among our method, MoPa, and TMR in Figure 5. While baseline methods retrieve basic actions like "move the foot", they fail to capture specific semantic details such as "switch the foot". In contrast, our method precisely retrieves motions that fully align with the complex textual description depicting fine-grained actions. It demonstrates our method's effectiveness in aligning intricate motions with texts.

## 6 CONCLUSION

We introduce WaMo, a novel wavelet-based framework that significantly enhances Text-Motion Retrieval (TMR) through comprehensive multi-frequency trajectory analysis. WaMo addresses key challenges in TMR through three key modules. Trajectory Wavelet Decomposition fully captures multi-scale frequency information. Trajectory Wavelet Reconstruction ensures the preservation of spatial-temporal information. Disordered Motion Sequence Prediction improves the learning of temporal structures. Experimental results on HumanML3D and KIT-ML datasets indicate that WaMo outperforms existing methods by substantial margins.

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

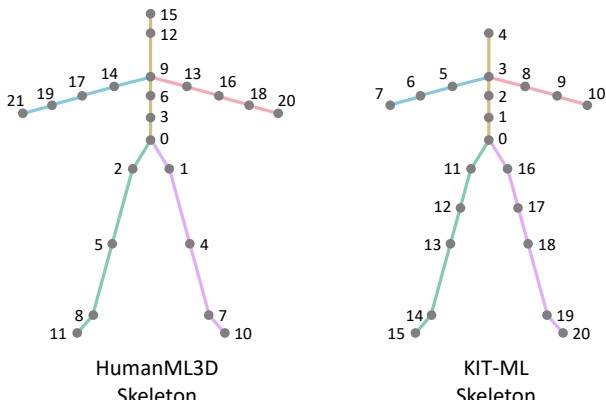

Figure 6: The human body skeletons of HumanML3D and KIT-ML datasets, respectively.

## A  MORE DATASET DETAILS

**HumanML3D** (Guo et al., 2022) contains 14,616 motions and 44,970 textual descriptions. The vocabulary has 5,371 unique words. The motion sequences are down-sampled to 20 FPS. The maximum length is set to 10 seconds, and longer sequences are truncated. The skeleton is shown in Figure 6, which has 22 joints. To enrich the dataset, motions and their corresponding text descriptions are augmented by mirroring left and right. The dataset is split into 80%, 5%, and 15% for training, validation, and test sets, respectively.

**KIT-ML** (Plappert et al., 2016) consists of 3911 motions and 6278 textual descriptions, with a vocabulary of 1,623 unique words. All motion sequences are down-sampled to 12.5 FPS. Each motion is annotated by 1 to 4 descriptions. On average, each textual description contains 8 words. The skeleton is illustrated in Figure 6, which has 21 joints. The dataset is divided into 80% for training, 5% for validation, and 15% for testing, respectively.

## B  MORE IMPLEMENTATION DETAILS

For the text encoder, we follow MoPa (Yu et al., 2024) to adopt pre-trained DistilBERT (Sanh et al., 2019) with the "distilbert-base-uncased" checkpoint[1]. In the Trajectory Wavelet Decomposition module, the learnable Stationary Wavelet Transform (SWT) is initialized by the db1 wavelet. In the Intra- and Inter-Frequency Attention module, each Transformer encoder has one standard Transformer layer (Vaswani et al., 2017). All Transformer layers have an expansion rate of 4 and are equipped with 4 multi-head attention heads. We adopt GELU (Hendrycks & Gimpel, 2016) as the activation function. For the 1D convolutions, the kernel sizes of low-frequency and high-frequency vectors are set to 7 and 3, respectively. Following prior work (Yu et al., 2024), the maximum length of motion sequences is set to 224, longer sequences are truncated, and shorter sequences are padded.

In all experiments, the model is trained for 100 epochs with a batch size of 128 on an NVIDIA RTX 3090 GPU with 24GB memory. To optimize the model, we adopt the Adam optimizer (Kingma & Ba, 2015) with a learning rate of 1e-4 and a cosine annealing schedule. We run experiments multiple times with different random seeds using the PyTorch-2.0.1 library (Paszke et al., 2017).

## C  TRAINING LOSS DETAILS

In the training process, our proposed method is optimized by the loss of each module:

$$\mathcal{L} = \mathcal{L}_{nce} + \lambda_{rec}\mathcal{L}_{rec} + \lambda_{dmsp}\mathcal{L}_{dmsp}, \tag{9}$$

where $\lambda_{rec}$ and $\lambda_{dmsp}$ are hyperparameters to balance losses. In the experiments, $\lambda_{rec}$ and $\lambda_{dmsp}$ are set to 5 and 1 on both datasets, respectively. We further report ablation studies on these loss weights in Section D.2.

---

[1]https://huggingface.co/distilbert/distilbert-base-uncased

| Method | Venue | Text-to-Motion | | | | | | Motion-to-Text | | | | | | Rsum↑ |
|---|---|---|---|---|---|---|---|---|---|---|---|---|---|---|
| | | R@1↑ | R@2↑ | R@3↑ | R@5↑ | R@10↑ | MedR↓ | R@1↑ | R@2↑ | R@3↑ | R@5↑ | R@10↑ | MedR↓ | |
| T2M | CVPR'22 | 52.48 | 71.05 | 80.65 | 89.66 | 96.58 | 1.39 | 52.00 | 71.21 | 81.11 | 89.87 | 96.78 | 1.38 | 781.39 |
| TEMOS | ECCV'22 | 40.49 | 53.52 | 61.14 | 70.96 | 84.15 | 2.33 | 39.96 | 53.49 | 61.79 | 72.40 | 85.89 | 2.33 | 623.79 |
| MotionCLIP | ECCV'22 | 46.24 | 60.25 | 68.93 | 80.47 | 91.35 | 1.88 | 44.76 | 56.81 | 65.22 | 77.83 | 90.19 | 2.03 | 682.05 |
| TMR | ICCV'23 | 67.16 | 81.32 | 86.81 | 91.43 | 95.36 | 1.04 | 67.97 | 81.20 | 86.35 | 91.70 | 95.27 | 1.03 | 844.57 |
| LAVIMO | CVPR'24 | 68.58 | 81.04 | 85.02 | 88.77 | 92.58 | 1.01 | 68.64 | 81.06 | 85.52 | 88.76 | 92.82 | 1.01 | 832.79 |
| MoPa | CVPR'24 | 71.61 | 85.81 | 90.02 | 94.35 | 97.69 | **1.00** | 72.11 | 85.26 | 90.21 | 94.44 | 97.76 | **1.00** | 879.26 |
| LaMP | ICLR'25 | 67.18 | 81.90 | 87.04 | 92.00 | 95.73 | - | 68.02 | 82.10 | 87.50 | 92.20 | 96.90 | - | 850.57 |
| MonSTeR | ICCV'25 | 67.38 | 82.96 | 88.89 | 93.75 | 97.45 | - | 67.91 | 82.64 | 88.14 | 93.06 | 97.43 | - | |
| **WaMo (Ours)** | - | **73.46** | **87.94** | **91.81** | **95.34** | **97.91** | **1.00** | **74.01** | **87.24** | **91.68** | **95.43** | **97.94** | **1.00** | **892.76** |

Table 9: **Comparison results on HumanML3D (Guo et al., 2022) with the "Small Batches" evaluation protocol.** The best results are shown in **bold** and the second-best outcomes are underlined. We achieve state-of-the-art results across all metrics.

| Method | Venue | Text-to-Motion | | | | | | Motion-to-Text | | | | | | Rsum↑ |
|---|---|---|---|---|---|---|---|---|---|---|---|---|---|---|
| | | R@1↑ | R@2↑ | R@3↑ | R@5↑ | R@10↑ | MedR↓ | R@1↑ | R@2↑ | R@3↑ | R@5↑ | R@10↑ | MedR↓ | |
| T2M | CVPR'22 | 42.25 | 62.62 | 75.12 | 87.50 | 96.12 | 1.88 | 39.75 | 62.75 | 73.62 | 86.88 | 95.88 | 1.95 | 722.49 |
| TEMOS | ECCV'22 | 43.88 | 58.25 | 67.00 | 74.00 | 84.75 | 2.06 | 41.88 | 55.88 | 65.62 | 75.25 | 85.75 | 2.25 | 652.26 |
| MotionCLIP | ECCV'22 | 41.29 | 55.38 | 69.50 | 78.83 | 90.12 | 1.73 | 39.55 | 52.07 | 68.13 | 77.94 | 90.85 | 2.16 | 663.66 |
| TMR | ICCV'23 | 49.25 | 69.75 | 78.25 | 87.88 | 95.00 | 1.50 | 50.12 | 67.12 | 76.88 | 88.88 | 94.75 | 1.53 | 757.88 |
| LAVIMO | CVPR'24 | 58.10 | 77.80 | 86.34 | 93.08 | 96.47 | 1.08 | 60.23 | 77.52 | 86.44 | 93.22 | 95.87 | 1.20 | 825.07 |
| MoPa | CVPR'24 | 53.55 | 71.30 | 79.82 | 88.92 | 96.29 | 1.36 | 54.54 | 72.15 | 79.68 | 89.35 | 96.11 | 1.31 | 781.71 |
| LaMP | ICLR'25 | 52.50 | 74.80 | 84.70 | 92.70 | 97.60 | - | 54.00 | 75.30 | 84.40 | 92.20 | 97.60 | - | 805.80 |
| **WaMo (Ours)** | - | **62.12** | **82.88** | **89.63** | **94.00** | **97.73** | **1.04** | **63.50** | **78.25** | **88.63** | **94.25** | **97.88** | **1.18** | **848.87** |

Table 10: **Comparison results on KIT-ML (Plappert et al., 2016) with the "Small Batches" evaluation protocol.** The best results are shown in **bold** and the second-best are underlined. We achieve state-of-the-art results across all metrics.

## D   MORE EXPERIMENTAL RESULTS

### D.1   DIFFERENT EVALUATION PROTOCOL

In the main text, we have presented retrieval results using the standard retrieval evaluation protocol. Prior works (Yu et al., 2024) also adopt another evaluation protocol, namely "Small Batches", which is proposed by (Guo et al., 2022). The method involves randomly sampling batches comprising 32 motion-text pairs, and subsequently computing the average retrieval metric. Although this approach introduces randomness, it establishes a comparative benchmark. Notably, the gallery size of 32 remains relatively smaller than the standard retrieval evaluation protocol, thereby making it a less challenging scenario.

Concretely, in Table 9 and Table 10, we report retrieval results under the "Small Batches" evaluation protocol on HumanML3D and KIT-ML, respectively. The compared baselines include: T2M (Guo et al., 2022), TEMOS (Petrovich et al., 2022), MotionCLIP (Tevet et al., 2022), TMR (Petrovich et al., 2023), LAVIMO (Yin et al., 2024), MoPa (Yu et al., 2024), and LaMP (Li et al., 2025). The results are sourced from MoPA, LAVIMO, and the respective original papers. Note that we achieve state-of-the-art results across all metrics on both datasets, demonstrating the superiority of our method in aligning 3D human motions and text descriptions.

### D.2   LOSS WEIGHTS

In Figure 7, we illustrate the sensitivity of our model to loss weights $\lambda_{rec}$ and $\lambda_{dmsp}$. It is found that our loss weight choices ($\lambda_{rec}$=5 and $\lambda_{dmsp}$=1) obtain the best results on both datasets. Moreover, the results demonstrate consistent performance with limited variations across different selections of $\lambda_{rec}$ and $\lambda_{dmsp}$. It indicates the robustness and generalization of our method across diverse hyperparameter settings.

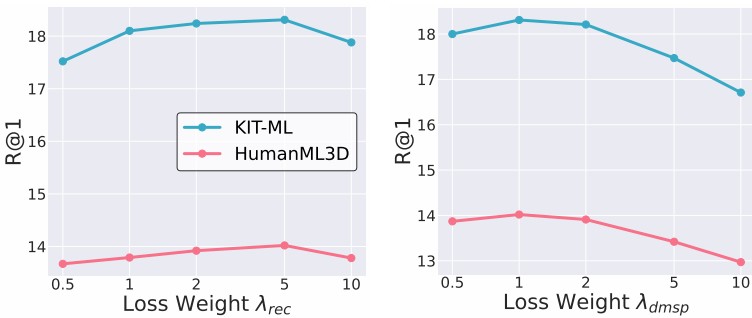

Figure 7: Ablation study on loss weights $\lambda_{rec}$ and $\lambda_{dmsp}$.

| Wavelet | HumanML3D | | | KIT-ML | | |
|---|---|---|---|---|---|---|
| | R@1↑ | R@2↑ | R@3↑ | R@1↑ | R@2↑ | R@3↑ |
| db1 | **14.02** | **17.58** | **25.51** | **18.31** | **24.82** | **34.46** |
| db2 | 13.84 | 17.32 | 24.61 | 18.11 | 23.11 | 33.13 |
| db4 | 13.48 | 16.96 | 24.11 | 17.95 | 23.14 | 31.81 |
| db8 | 13.87 | 17.39 | 24.53 | 18.00 | 23.98 | 30.24 |
| db12 | 13.41 | 16.72 | 23.94 | 17.47 | 23.83 | 30.30 |
| bior3.1 | 13.76 | 16.80 | 24.28 | 18.10 | 24.10 | 32.41 |

Table 11: Ablation study on the wavelet initialization in text-to-motion retrieval. The best results are shown in **bold** and the second-best outcomes are underlined.

### D.3 WAVELET INITIALIZATION

In Table 11, we assess the impact of different wavelet initialization settings in the learnable SWT. We validate widely-adopted wavelet families {db1, db2, db4, db8, db12, bior3.1}. The db1 (Haar) wavelet achieves the optimal results on both datasets, primarily due to its effectiveness in capturing short and abrupt motions. Besides, the db1 wavelet achieves the best performance across both the HumanML3D (diverse daily actions) and KIT-ML (locomotion motions) datasets. This cross-dataset consistency demonstrates the robustness and generalization ability.

### D.4 EFFICIENCY COMPARISON

In Table 12, we report the computational overhead and text-to-motion retrieval accuracy of MoPa (Yu et al., 2024), our base model without proposed modules (i.e., wavelet transforms and Disordered Motion Sequence Prediction), and our method. The training process is equipped with early stopping.

Compared to the base model, although our method leads to increased parameter size, memory usage, and training/inference time, it also significantly improves the retrieval accuracy. Compared to MoPa, we achieve better retrieval accuracy with faster inference speed and fewer parameters. In particular, although our method increases training time in one epoch, the total training time only slightly increases. This is due to our method converging using fewer epochs, indicating better learning efficiency.

### D.5 RESULTS ON LARGER AND MORE DIVERSE DATASET

We further validate our method on the Motion-X++ dataset (Zhang et al., 2025). Compared to HumanML3D and KIT-ML, it is a larger dataset, containing longer motion sequences with longer and noisier text annotations (generated by LLMs) in more diverse scenarios. We report results of the previous SOTA method MoPa (Yu et al., 2024), our base model without proposed modules (i.e., wavelet transforms and Disordered Motion Sequence Prediction), and our full setup in Table 13.

| Method | Parameters | Memory (Train) | Memory (Inference) | Train (Batch) | Train (Total) | Inference | Text-to-Motion | | |
|---|---|---|---|---|---|---|---|---|---|
| | | | | | | | R@1↑ | R@2↑ | R@3↑ |
| MoPa | 152.59M | 19450M | 1212 MB | 99.61s | 8019.34s | 11.27s | 10.8 | 14.98 | 20 |
| Base Model | **80.29M** | **18228M** | **672 MB** | **90.26s** | **6475.19s** | **7.21s** | 9.14 | 13.65 | 18.72 |
| **WaMo (Ours)** | 84.34M | 21258M | 812 MB | 156.64s | 6749.12s | 10.46s | **14.02** | **17.58** | **25.51** |

Table 12: Efficiency comparison between MoPa (Yu et al., 2024), the base model without the proposed module, and our method on HumanML3D. The metrics follow the standard evaluation protocol as in the main text.

| Method | Text-to-Motion | | | Motion-to-Text | | |
|---|---|---|---|---|---|---|
| | R@1↑ | R@2↑ | R@3↑ | R@1↑ | R@2↑ | R@3↑ |
| MoPa | 15.65 | 22.64 | 27.59 | 15.63 | 23.08 | 27.82 |
| Base Model | 19.34 | 27.54 | 33.14 | 20.58 | 29.09 | 34.66 |
| **WaMo (Base Model + Ours)** | **24.11** | **33.11** | **39.20** | **24.19** | **34.84** | **40.85** |

Table 13: Comparison results on Motion-X++ (Zhang et al., 2025). The base model is trained without the proposed modules. The best results are shown in **bold**.

In particular, our method significantly outperforms MoPa. This is due to that MoPa pools joints into five body parts (torso, arms, legs), leading to significant information loss in dense skeletons (i.e., Motion-X++ with 52 joints compared to HumanML3D with 22 joints). In contrast, our method applies wavelet decomposition to the trajectory of every individual joint, preserving the fine-grained motion details and leading to better retrieval accuracy.

## D.6    T-SNE VISUALIZATION

To evaluate the semantic alignment quality, in Figure 8, we visualize the latent space using t-SNE on a randomly sampled subset of text-motion pairs from the KIT-ML test set. As shown in Figure 8 (a), the base model produces poorly aligned feature distributions. Motion and text embeddings are located in distinct regions. This separation indicates inadequate cross-modal correspondence. In contrast, our method demonstrates improved alignment, as demonstrated in Figure 8 (d). Motion and text embeddings are tightly interleaved in clusters, indicating effective cross-modal alignment. There are also clear separations between clusters, representing the discrimination of distinct semantics. It verifies the effectiveness of our method in aligning motions and text.

We also present t-SNE visualizations at different decomposition levels $S$ in Figure 8 (b-e). The findings are consistent with the quantitative results in Section 5.3.3. As $S$ increases from 1 to 3, text and motion embeddings become more tightly interleaved within clusters, indicating better cross-modal alignment. It demonstrates that deeper decomposition captures richer temporal hierarchies, improving the text-motion alignment. When $S = 4$, the alignment quality slightly drops compared to $S = 3$, with clusters becoming less compact. This is due to that excessive decomposition may introduce noisy and motion-irrelevant features, limiting the precise matching with text semantics.

## D.7    LEARNED WAVELET FILTERS

We visualize the scaling function $\phi(t)$ (low-pass) and wavelet function $\psi(t)$ (high-pass) before and after training in Figure 9. The wavelet filters are initialized with the db1 wavelet, which consists of rigid box and step functions. After training on human motion data, the filters adapt significantly. The learned scaling function exhibits a non-uniform, decaying structure. It allows the model to better capture the smooth, low-frequency dynamics inherent in motion trajectories, instead of a simple moving average in standard db1 wavelets. The learned wavelet function transforms into a waveform with more oscillations. It captures complex high-frequency components of motions (e.g., sudden limb accelerations), which standard db1 wavelets might over-smooth. The visual comparison

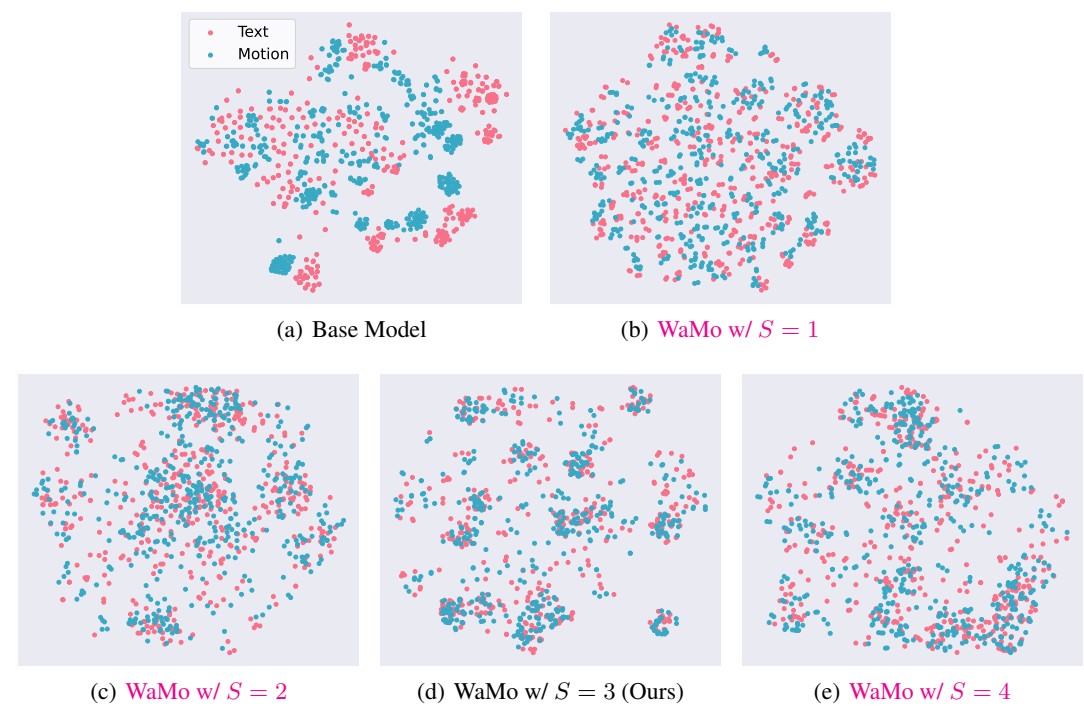

Figure 8: t-SNE visualizations (van der Maaten & Hinton, 2008) on KIT-ML. (a) is the base model trained without the proposed modules. (b-e) shows the model trained with full setup on different decomposition levels $S$.

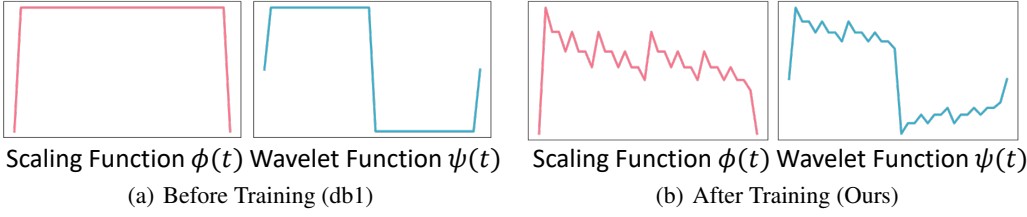

Figure 9: Visualizations of wavelet filters before and after training. (a) is initialized with the db1 wavelet. (b) shows the learned wavelet filters after training.

indicates that our learnable filters successfully adapt to the specific spatial-temporal characteristics of 3D human motion.

To quantitatively verify the effectiveness of learnable wavelets, we compare our method with a variant using the fixed db1 wavelet. As shown in Table 14, using learnable wavelets leads to better retrieval accuracy across all metrics on both datasets.

## D.8 GENERALIZATION OF DISORDERED MOTION SEQUENCE PREDICTION

To demonstrate the effectiveness of Disordered Motion Sequence Prediction (DMSP), we first integrate it into two existing models: T2M (GRU-based) (Guo et al., 2022) and TMR (Transformer-based) (Petrovich et al., 2023). As shown in Table 15, DMSP consistently achieves performance improvements across different model architectures.

To further demonstrate the generalization ability of DMSP in other motion-related tasks, we further apply our method on two additional tasks: Zero-shot Motion Classification and Human Interaction Recognition. As shown in Table 16 and 17, DMSP leads to performance improvements in all metrics.

| Setting | HumanML3D | | | KIT-ML | | |
|---|---|---|---|---|---|---|
| | R@1↑ | R@2↑ | R@3↑ | R@1↑ | R@2↑ | R@3↑ |
| WaMo w/ fixed wavelet | 12.56 | 15.67 | 22.64 | 16.37 | 23.29 | 31.14 |
| **WaMo w/ learnable wavelet (Ours)** | **14.02** | **17.58** | **25.51** | **18.31** | **24.82** | **34.46** |

Table 14: Ablation study on learnable wavelet filters.

| Setting | HumanML3D | | | KIT-ML | | |
|---|---|---|---|---|---|---|
| | R@1↑ | R@2↑ | R@3↑ | R@1↑ | R@2↑ | R@3↑ |
| T2M | 1.80 | 3.42 | 4.79 | 3.37 | 6.99 | 10.84 |
| **T2M w/ DMSP** | **3.45** | **5.12** | **7.31** | **5.97** | **9.46** | **14.23** |
| TMR | 5.68 | 10.59 | 14.04 | 7.23 | 13.98 | 20.36 |
| **TMR w/ DMSP** | **8.19** | **12.75** | **16.87** | **9.64** | **16.78** | **23.47** |

Table 15: Results of the Disordered Motion Sequence Prediction module on existing methods.

# E  APPLICATIONS

To demonstrate the generalization ability of our method, following MoPa (Yu et al., 2024), we further apply our method on two additional tasks: Zero-shot Motion Classification and Human Interaction Recognition.

## E.1  ZERO-SHOT MOTION CLASSIFICATION

We adopt the BABEL 60-class benchmark (Punnakkal et al., 2021), which contains 10892 sequences, and 20% of them are used as the test set. The motions are processed following the same procedure as HumanML3D. We directly apply our model trained on HumanML3D to the test set. The action labels in BABEL are used as "A person action" for classification. We then calculate the cosine similarity between a given motion and all 60 action labels. The action label with the highest similarity is taken as the final classification category. The Top-1 and Top-5 accuracy are shown in Table 16, where our method outperforms TMR (Petrovich et al., 2023) and MoPa.

## E.2  HUMAN INTERACTION RECOGNITION

We use the InterHuman Dataset (Liang et al., 2024) to demonstrate that our method can also be applied to multi-person motion recognition, beyond single-person motion recognition. InterHuman consists of diverse interactions between two individuals, which are split into 6222 samples for training and 1557 for testing. Following MoPa, we adopt a shared motion encoder for each individual's motion. The motion features are then concatenated, followed by a projection layer. The results are shown in Table 17, where our method outperforms TMR and MoPa.

# F  MORE QUALITATIVE RESULTS

## F.1  FINE-GRAINED TEXT-TO-MOTION RETRIEVAL

In Figure 10, we present more comparative visualizations of fine-grained text-to-motion retrieval results among our method, MoPa (Yu et al., 2024), and TMR (Petrovich et al., 2023). The results indicate the effectiveness of our method in aligning fine-grained motions and text descriptions.

## F.2  WAVELET DECOMPOSITION OF HUMAN MOTION

In Figure 11 and 12, we present more visualizations of wavelet decomposition of human motion. These visualizations demonstrate how frequency-specific decomposition reveals the motion-text correspondence.

| Method | Top-1 Acc.↑ | Top-5 Acc.↑ |
|---|---|---|
| TMR | 30.13 | 41.52 |
| MoPa | 41.33 | 68.97 |
| WaMo w/o DMSP | 41.95 | 69.29 |
| **WaMo (Ours)** | **43.19** | **70.21** |

Table 16: Results of zero-shot motion classification. The best results are shown in **bold**.

| Method | Text-to-Motion | | | Motion-to-Text | | |
|---|---|---|---|---|---|---|
| | R@1↑ | R@5↑ | R@10↑ | R@1↑ | R@5↑ | R@10↑ |
| TMR | 5.38 | 15.64 | 24.40 | 5.13 | 15.26 | 25.65 |
| MoPa | 9.51 | 21.27 | 32.41 | 8.26 | 22.65 | 32.66 |
| WaMo w/o DMSP | 10.86 | 24.09 | 35.68 | 10.56 | 24.53 | 35.51 |
| **WaMo (Ours)** | **13.08** | **25.94** | **37.13** | **12.64** | **25.97** | **36.16** |

Table 17: Results of human interaction recognition. The best results are shown in **bold**.

## G  THE USE OF LARGE LANGUAGE MODELS

Large language models (LLMs) are adopted to polish the writing by improving sentence clarity, flow, and grammar.

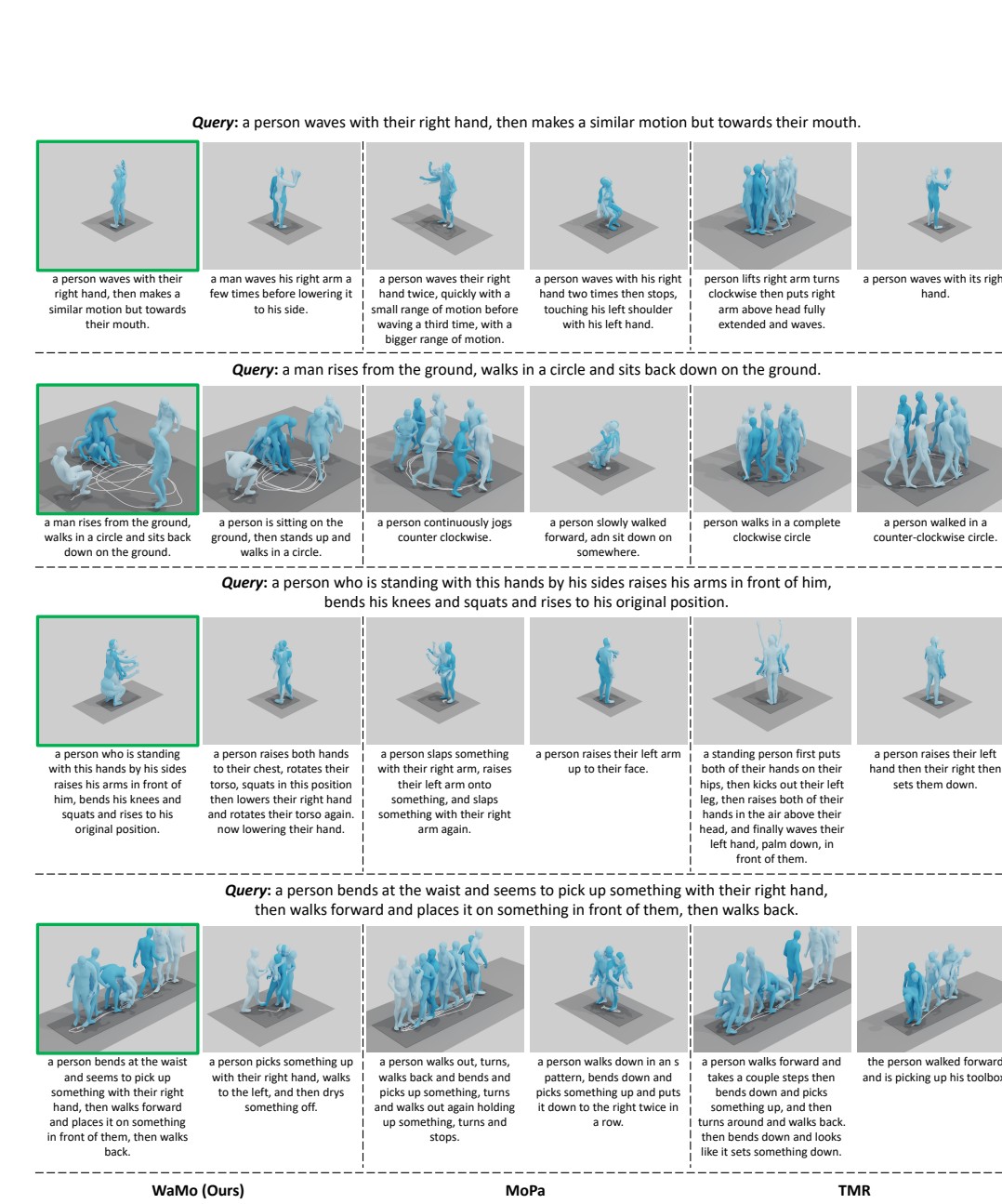

Figure 10: **Visualization comparisons of fine-grained text-to-motion retrieval results between our method, MoPa (Yu et al., 2024), and TMR (Petrovich et al., 2023) on HumanML3D.** The top-2 retrieval results are shown from left to right. The ground-truth motion is highlighted by a green box. Zoom in for better visibility.

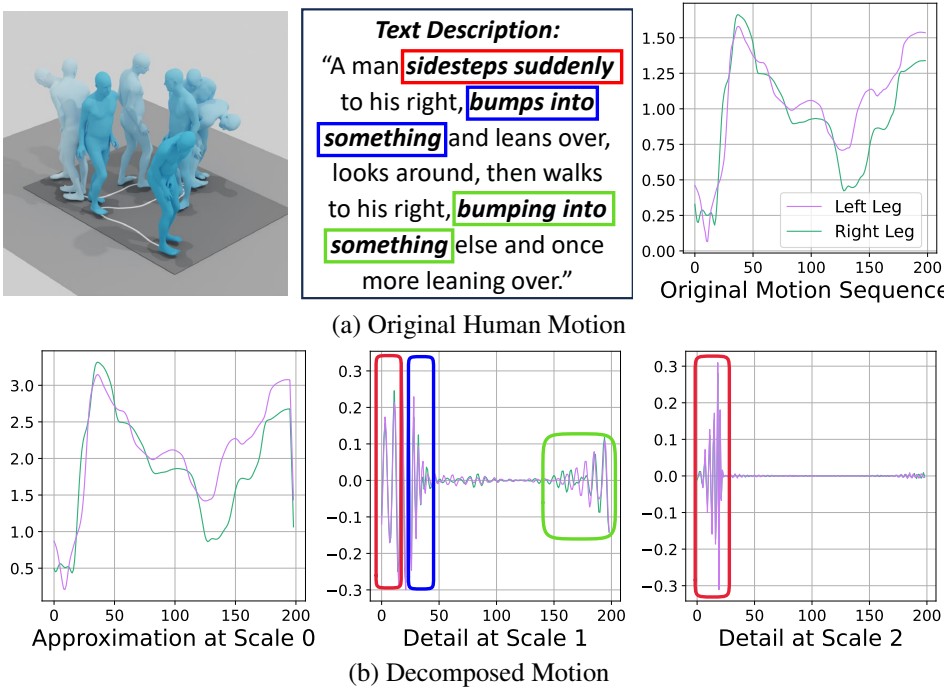

(a) Original Human Motion

(b) Decomposed Motion

Figure 11: **Wavelet decomposition of human motion signals.** For the original human motion trajectory of left and right legs (shown in the top-right subfigure, where the x-axis denotes time and the y-axis denotes the amplitude of movements), we perform wavelet decomposition across three distinct scales. The scale-0 waveform preserves the overall structure of the original motion sequence, showing general trends of the motion. The scale-1 waveform captures mid-high frequency features: the dense oscillations around frame 0 indicate the "sidesteps suddenly" (as in red boxes) and "bumps into something" (as in blue boxes) actions, while fluctuations after frame 150 reflect the "bumps again" movement (as in green boxes). Scale 2 highlights higher-frequency, further emphasizing the rapid "sidesteps suddenly" action at frame 0. These multi-scale representations reveal fine-grained motion semantics, aligning closely with textual descriptions.

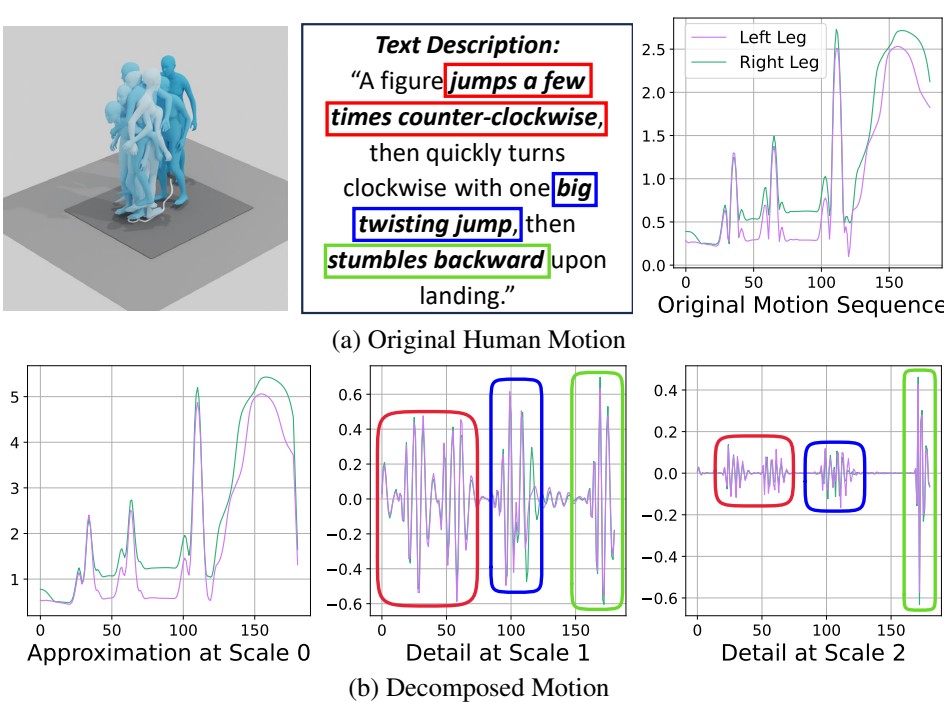

(a) Original Human Motion

(b) Decomposed Motion

Figure 12: **Wavelet decomposition of human motion signals.** For the original human motion trajectory of left and right legs (shown in the top-right subfigure, where the x-axis denotes time and the y-axis denotes the amplitude of movements), we perform wavelet decomposition across three distinct scales. The scale-0 waveform preserves the overall structure of the original motion sequence, showing general trends of the motion but lacking fine-grained action details. The scale-1 waveform captures mid-high frequency features, aligning peaks with the dynamic phases of actions. Early peaks correspond to the "jumps a few times counter-clockwise" action (as in red boxes), and a huge peak around frame 100 matches the "big clockwise twisting jump" movement (as in blue boxes). The subsequent fluctuations reflect the "stumbles backward" action (as in green boxes). Scale 2 focuses on high-frequency features, highlighting the short and abrupt "stumbles backward" action with higher amplitude. These multi-scale representations reveal fine-grained motion semantics, aligning closely with textual descriptions.

