# OpenReview forum: "WaMo: Wavelet-Enhanced Multi-Frequency Trajectory Analysis for Fine-Grained Text-Motion Retrieval"
_ICLR.cc/2026/Conference — Submitted to ICLR 2026_

### Official Review · Reviewer_3yLM · 2025-10-31

**Soundness:** 4
**Presentation:** 3
**Contribution:** 3
**Rating:** 8
**Confidence:** 3

**Summary:**

This paper proposes a wavelet-based motion representation that incorporates multi-frequency modeling to capture both global structure and local details in human motion. The use of an inverse wavelet reconstruction loss and a temporal reordering classification loss effectively regularizes the representation, encouraging stronger spatiotemporal awareness. Extensive experiments show significant improvements in text-motion retrieval over existing baselines, and comprehensive ablation studies confirm the contribution of each proposed component.

**Strengths:**

- The paper is clearly written and well-organized. The idea of integrating wavelet decomposition for motion representation is intuitive and well-grounded. The technical design is elegant and has been shown to be lightweight in supplemental experiments.

- The multi-scale modeling effectively captures both long-term structure and fine motion details, addressing a key weakness of prior models.

- The experimental evaluation is very comprehensive, with multiple baselines, ablations, and qualitative visualizations that support the main claims.

**Weaknesses:**

- The authors note that FFT-based methods struggle to capture local motion details. However, some prior works have explored multi-level or part-level feature modeling within FFT frameworks, demonstrating benefits for motion representation. A deeper discussion of these approaches and, if possible, an ablation comparing against such variants would further strengthen the paper’s analysis.
    - Starke, Sebastian, et al. "Local motion phases for learning multi-contact character movements." ACM Transactions on Graphics 39.4 (2020).
    - Starke, Sebastian, Ian Mason, and Taku Komura. "Deepphase: Periodic autoencoders for learning motion phase manifolds." ACM Transactions on Graphics (ToG) 41.4 (2022): 1-13.
    - Wan, Weilin, et al. "Diffusionphase: Motion diffusion in frequency domain." arXiv preprint arXiv:2312.04036 (2023).
- Additional visualizations similar to Figure 1 would be helpful for interpreting the effectiveness of the wavelet formulation for modeling fine-grained details. Moreover, videos may be clearer than figures, for example in Figure 1, the two-hand interaction is difficult to see in the human motion image.

**Questions:**

- What value is used for J? Does it correspond to the total number of joints in the dataset, or is it based on a part-level grouping? Figure 2 appears to suggest ( J = 5 ), where “l-arm,” “r-arm,” etc., denote body parts that each aggregate multiple joints. If so, how is the trajectory of such a grouped body part defined?

- How are the temporal segments divided for DMSP? Are they evenly split or determined by motion characteristics?

- What is the reconstruction quality of TWR across different levels of wavelet decomposition? It would be interesting to quantify the degree of information loss and discuss whether this loss meaningfully affects retrieval performance.

- The t-SNE visualization in Figure 8 (b) seems to be sparser (i.e., fewer dense clusters). Could the authors provide insights into why this occurs? It would also be interesting to visualize how the t-SNE distribution changes across different levels of wavelet decomposition.

---

> ### Author Response · Authors · 2025-11-22
> **Response to Reviewer 3yLM (Part 1 of 2)**
>
> Thank you for your valuable time and thoughtful feedback on our manuscript, which has greatly helped us improve it. We address each of your points below.
>
> > **Weakness 1 - Discussion about FFT-based methods**
>
> We thank the reviewer for highlighting these relevant works. We have discussed them in the revised paper and clarified the distinctions below.
>
> These works [1,2,3] primarily leverage frequency analysis for periodic locomotion or global phase manifolds. While effective for continuous motion generation, standard FFT captures global frequency components but struggles to handle non-stationary signals [4], such as short and abrupt motions (e.g., sudden acceleration of a limb), which are essential for motion retrieval.
>
> In contrast, we adopt Stationary Wavelet Transform (SWT), which preserves the temporal location of multi-scale features. This is critical for TMR, where text descriptions usually align with specific motion segments rather than global periodic cycles.
>
> We present ablation studies in Table 4 (Row 2 vs. Row 5) of our paper for ablation comparing FFT-based modeling against our wavelet decomposition. As shown in the table below, the FFT-based variant underperforms WaMo. The performance gap suggests that while FFT captures global signals, the shift-invariant, time-preserving nature of SWT in WaMo is superior for fine-grained text-motion alignment.
>
> || HumanML3D ||| KIT-ML|||
> | - | - | - | - | - | - | - |
> || R@1| R@2 | R@3 | R@1| R@2| R@3 |
> | WaMo w/ FFT| 12.10| 15.44| 22.07| 15.98| 23.21| 30.62|
> | **WaMo w/ SWT (Ours)** | **14.02** | **17.58** | **25.51** | **18.31** | **24.82** | **34.46** |
>
> > **Weakness 2 - Additional visualizations similar to Figure 1**
>
> We have provided additional visualizations in the revised paper. Please refer to Section F.2 in the revised appendix. The videos can be found in the supplementary material.
>
> ---
>
> References
>
> [1] Starke, Sebastian, et al. "Local motion phases for learning multi-contact character movements." ACM Transactions on Graphics 39.4 (2020).
>
> [2] Starke, Sebastian, Ian Mason, and Taku Komura. "Deepphase: Periodic autoencoders for learning motion phase manifolds." ACM Transactions on Graphics (ToG) 41.4 (2022): 1-13.
>
> [3] Wan, Weilin, et al. "Diffusionphase: Motion diffusion in frequency domain." arXiv preprint arXiv:2312.04036 (2023).
>
> [4] Alaa, Ahmed, Alex James Chan, and Mihaela van der Schaar. "Generative time-series modeling with fourier flows." International Conference on Learning Representations. 2021.

---

> ### Author Response · Authors · 2025-11-22
> **Response to Reviewer 3yLM (Part 2 of 2)**
>
> > **Question 1 - What value is used for J?**
>
> We adopt the total number of joints in the datasets, J is set to 22 and 21in HumanML3D and KIT-ML, respectively. Instead of part-level grouping, we apply wavelet transforms to every joint to capture the fine-grained motion dynamics of every individual joint.
>
> To simplify the visualizations in Figure 2, we use kinematic chains of skeletons (Figure 6) to group joints into five body parts (torso, arms, legs), pooling the xyz coordinates of corresponding joints.
>
> > **Question 2 - How are the temporal segments divided for Disordered Motion Sequence Prediction (DMSP)?**
>
> The temporal segments for DMSP are evenly split. They are determined by the Group Number $\lambda_g$. For a motion sequence with T frames, each temporal segment contains T/$\lambda_g$ frames. Figure 4 indicates that the method's performance is stable across different $\lambda_g$ values. The stability is due to the model's strong ability to learn temporal dependencies through the sequence reordering task. Even when segment sizes vary, the DMSP module successfully captures the general temporal flow of motions, preventing dependence on specific segment sizes.
>
> > **Question 3 - What is the reconstruction quality of TWR across different levels of wavelet decomposition?**
>
> We report the Fréchet Inception Distance (FID) to measure the reconstruction quality of TWR across different wavelet decomposition levels ($S$). The results are shown in the table below.
>
> || HumanML3D |||| KIT-ML||||
> | - | - | - | - | - | - | - | - | - |
> | $S$| FID↓| R@1| R@2| R@3| FID↓| R@1| R@2| R@3|
> | 1| 0.049| 11.93| 15.75| 22.21| 0.152| 15.66| 22.85| 30.26|
> | 2| 0.032| 12.81| 16.39| 23.16| 0.127| 16.99| 23.73| 30.96|
> | **3** | **0.018** | **14.02** | **17.58** | **25.51** | **0.085** | **18.31** | **24.82** | **34.46** |
> | 4| 0.026| 13.84| 17.20| 24.52| 0.112| 18.19| 24.06| 31.33|
>
> When the decomposition level increases from $S=1$ to $S=3$, FID decreases significantly, indicating less information loss. The improvement in motion reconstruction also enhances retrieval accuracy, as the model captures more complex temporal hierarchies. However, at $S=4$, there is a slight increase in FID and a drop in retrieval metrics. This suggests that excessive decomposition may introduce noise and motion-irrelevant features, limiting both reconstruction quality and text-motion matching.
>
> > **Question 4.1 - Why does the t-SNE visualization of WaMo seem to be sparser?**
>
> In the baseline model (Figure 8 (a)), text and motion features with similar semantics fail to align. They occupy distinct regions in the latent space. This separation leads to more clusters. In contrast, our method (Figure 8 (d) in the revised paper) successfully pulls semantically related text and motion features together. As they become tightly interleaved, the previously separated clusters merge into unified, semantically dense clusters.
>
> > **Question 4.2 - How does the t-SNE distribution change across different levels of wavelet decomposition?**
>
> We present t-SNE visualizations at different decomposition levels ($S$) in Section D.6 of the revised appendix. The findings are consistent with the quantitative results in the response to Question 3. As $S$ increases from 1 to 3, text and motion embeddings become more tightly interleaved within clusters, indicating better cross-modal alignment. It demonstrates that deeper decomposition captures richer temporal hierarchies, improving the text-motion alignment. When $S=4$, the alignment quality slightly drops compared to $S=3$, with clusters becoming less compact. This is due to that excessive decomposition may introduce noisy and motion-irrelevant features, limiting the precise matching with text semantics.

---

### Official Review · Reviewer_1jTc · 2025-10-31

**Soundness:** 3
**Presentation:** 3
**Contribution:** 2
**Rating:** 4
**Confidence:** 5

**Summary:**

This paper proposes WaMo, a framework for text-to-motion retrieval that models human motion at multiple temporal scales. The key idea is to decompose joint trajectories into frequency bands using wavelets, reconstruct them, and also predict the correct temporal order of shuffled motion segments. This design aims to capture both slow, global trends and fast, local movements, improving alignment between motion and text descriptions. Experiments on HumanML3D and KIT-ML show improved retrieval performance compared with prior methods.

**Strengths:**

•	Uses multi-frequency decomposition to better represent different motion dynamics across time scales.

•	The DMSP task encourages the model to understand temporal structure, which may help fine-grained text-motion alignment.

•	Empirical gains on standard datasets demonstrate practical effectiveness.

**Weaknesses:**

1.	**Motivation needs further clarification.**
The motivation presented in Figure 1 is not very representative. From the figure, the three frequency scales show rather flat trajectories between frames 50–100, followed by sharp changes around 150. However, it remains unclear what these low- and high-frequency components actually correspond to in terms of specific human motions. The paper would benefit from deeper analysis and interpretation of the relationship between motion dynamics and their frequency-domain characteristics.
2.	**Limited novelty in core ideas.**
The concepts of wavelet and frequency decomposition have been explored in several prior works related to human motion (e.g., WaveAR, MotionWavelet, HumanMAC, LearnTrajDep). The authors should better articulate what is unique about applying such ideas to text-to-motion retrieval (TMR). What task-specific challenges or opportunities make the wavelet-based approach particularly suitable here?
3.	**Misleading claim about prior work.**
The paper states that existing methods “indiscriminately process human motion across different parts and moments.” This claim is inaccurate. For example, MotionPatch (CVPR 2024) already incorporates patch-based motion modeling that captures local spatiotemporal structure and part-wise motion patterns. The discussion should acknowledge such advances and clarify how WaMo differs beyond this level.
4.	**Common idea for Disordered Motion Sequence Prediction (DMSP).**
The idea of shuffling and reconstructing temporal order is widely used in other domains such as video and image tasks (e.g., Jigsaw Puzzle). The authors should explain what is special or necessary about applying this strategy to motion data. Without a clearer task-specific motivation, DMSP appears to be a direct adaptation of a generic self-supervised idea.
5.	**Use of MoMask for evaluation.**
MoMask is primarily designed for text-to-motion generation. It is unclear how it is used here to evaluate text-to-motion retrieval performance. More explanation is needed on how the retrieval results are computed or adapted from the generation framework.
6.	**Computational overhead unaddressed.**
Frequency-based modeling introduces additional computation, yet the paper provides no discussion or quantitative benchmark of training/inference speed or memory usage. A comparison with existing lightweight TMR models would strengthen the empirical evaluation.
7.	**Limited dataset diversity.**
All experiments are conducted on HumanML3D and KIT-ML, which are relatively small, clean, and constrained. The generalization ability to larger datasets (e.g., Motion-X) remains unverified. Broader evaluation would be necessary to demonstrate robustness.
8.	**Unexpectedly poor LaMP results.**
The reported LaMP performance is substantially lower than what was shown in the original paper, where it slightly outperformed baseline TMR methods. Please double-check the evaluation protocol or implementation to ensure fairness and reproducibility.
9.	**Strong dependency on wavelet basis choice.**
Table 10 shows that performance is sensitive to the wavelet initialization (e.g., db1 achieves the best result). The paper should discuss why the Haar wavelet (db1) is particularly effective for human motion modeling and whether this suggests dataset-specific bias or inherent alignment between Haar filters and certain motion patterns.

**Questions:**

See Weaknesses

---

> ### Author Response · Authors · 2025-11-22
> **Response to Reviewer 1jTc (Part 1 of 4)**
>
> Thank you for your valuable time and thoughtful feedback on our manuscript, which has greatly helped us improve it. We address each of your points below.
>
> > **Weakness 1 - Further clarification of our motivation presented in Figure 1**
>
> Thank you for the comment. To further clarify our motivation presented in Figure 1, we have followed your suggestion and provided a more detailed explanation of how the low- and high-frequency components correspond to specific human motion patterns.
>
> The Scale 0 **(low-frequency)** waveform preserves the overall structure of the original motion sequence. Since this scale filters out rapid and high-frequency dynamics, the trajectory between frames 50–100 is relatively flat. It reflects the stable position of the arms, while higher scales capture more specific action details.
>
> In the Scale 1 **(mid-high frequency)** waveform, the relatively sharper fluctuations around frames 50–100 (highlighted by the red box) correspond to the "wipe it off using right hand" action. It is a short and rapid local movement (the right hand quickly swipes the surface of the object), which generates high-frequency signals.
>
> In the Scale 2 **(higher frequency)** waveform, the fluctuations (highlighted by the blue box) around frame 150 correspond to the "put it down in his left hand" action. It involves fine-grained two-handed interactions. The right hand releases the object and the left hand catches it, which are rapid, small-range movements. This short and abrupt interaction generates higher-frequency signals than the stable and periodic single-hand 'wiping' action. Therefore, it is captured in the higher-scale (Scale 2) frequency component.
>
> We hope that our detailed analysis and explanation of the relationship between motion dynamics and their frequency characteristics could further clarify the motivation of our work.
>
> > **Weakness 2 - Novelty of WaMo**
>
> Thank you for your comment. We are happy to further clarify the novelty of WaMo. The discussions have been incorporated into the revised paper.
>
> While frequency decomposition has been used in single-modal motion prediction and generation [1-4], we are the first to introduce multi-frequency analysis specifically for cross-modal Text-Motion Retrieval (TMR). The novelty of WaMo for TMR lies in semantic alignment. Both text queries and motions inherently contain multi-granular information, ranging from global trends (low-frequency) to fine-grained details (high-frequency). WaMo employs wavelet decomposition not for smoothing or prediction, but to explicitly decompose these motion components, enabling precise alignment with the corresponding textual semantics.
>
> Besides, HumanMAC [1] and LearnTrajDep [2] use the Discrete Cosine Transform (DCT) to capture global dynamics for motion generation and prediction, but they discard high-frequency components to smooth human motion. However, in TMR, high-frequency details are usually semantically significant features (e.g., short and abrupt motion). Instead of smoothing motions, WaMo employs wavelets to extract these discriminative features to match specific semantics in the text query.
>
> WaveAR [3] and MotionWavelet [4] adopt wavelets to model non-stationary dynamics for motion prediction. However, they do not distinguish the fine-grained semantics of low- and high-frequency components. In TMR, the text queries describe motions at varying granularities, from global trends (e.g., walking) related to low-frequency components, to short and abrupt actions (e.g., sudden acceleration of a limb) related to high-frequency components. Therefore, we further capture both frequency-specific characteristics and their inter-dependencies through the Intra- and Inter-Frequency Attention module, facilitating fine-grained alignment with textual semantics.
>
> To quantitatively verify the effectiveness of our design, we compare WaMo with two variants: one replacing wavelets with DCT (corresponding to HumanMAC and LearnTrajDep), and one removing the Intra-Frequency Attention (corresponding to WaveAR and MotionWavelet). As shown in the table below, both modifications lead to clear performance drops, indicating the effectiveness of preserving high-frequency information and modeling intra- and inter-frequency semantics.
>
> || HumanML3D ||| KIT-ML|||
> | - | - | - | - | - | - | - |
> || R@1| R@2 | R@3 | R@1| R@2| R@3 |
> | WaMo w/ DCT| 12.69| 16.10| 22.84| 16.17| 23.96| 31.15|
> | WaMo w/o Intra-Frequency Attention | 12.82| 15.74| 23.36| 17.11| 22.71| 30.12|
> | **WaMo (Ours)**| **14.02** | **17.58** | **25.51** | **18.31** | **24.82** | **34.46** |
>
> ---
>
> References
>
> [1] HumanMAC: Masked Motion Completion for Human Motion Prediction, ICCV
>
> [2] Learning Trajectory Dependencies for Human Motion Prediction, ICCV
>
> [3] WaveAR: Wavelet-Aware Continuous Autoregressive Diffusion for Accurate Human Motion Prediction, NeurIPS
>
> [4] MotionWavelet: Human Motion Prediction via Wavelet Manifold Learning, arXiv

---

> ### Author Response · Authors · 2025-11-22
> **Response to Reviewer 1jTc (Part 2 of 4)**
>
> > **Weakness 3 - Discussion about prior work**
>
> We thank the reviewer for this comment. We would like to clarify that MoPa [1] uses a **coarse-grained** method to model spatiotemporal structure, limiting its effectiveness on dense skeletons. In contrast, WaMo captures **fine-grained** motion dynamics, leading to better preservation of complex kinematic details. The discussions have been incorporated into the revision.
>
> MoPa pools joints into **five body parts** (torso, arms, legs) to simplify the skeletons. In contrast, WaMo applies wavelet decomposition to the trajectory of **every** **individual joint**. The temporal dynamics of every joint are decomposed into multi-scale frequency bands to capture fine-grained motion characteristics, which coarse-grained part-level pooling may overlook.
>
> Besides, MoPa's joint pooling causes information loss for specific joints, while WaMo preserves the fine-grained information of every joint. It is particularly critical for dense skeletons (e.g., Motion-X++ with 52 joints compared to HumanML3D with 22 joints). On the Motion-X++ dataset, WaMo significantly outperforms MoPa, indicating the effectiveness of WaMo in capturing the complex dynamics of dense skeletons. Please refer to the response to Weakness 7 for results on Motion-X++.
>
> > **Weakness 4 - Novelty for Disordered Motion Sequence Prediction (DMSP)**
>
> We appreciate the reviewer’s feedback. While sequence reordering is common in 2D vision, its direct application to 3D human motion leads to performance degradations due to the fundamental difference in visual modality.
>
> Unlike 2D images/videos that contain dense static visual textures (e.g., background, object appearance) to provide context, 3D human motion is sparse and represented by kinematic dynamics (e.g., velocity and acceleration).
>
> Frame-level shuffling (as used in video tasks [2]) rearranges every frame across the entire sequence. Directly applying it to motion completely disrupts these local kinematic dependencies. It results in discontinuous, physically impossible actions rather than meaningful motion segments, confusing the encoder.
>
> In contrast, DMSP is specifically designed for the unique nature of motion. By dividing sequences into temporally coherent groups and only shuffling a small group of frames, DMSP preserves the local kinematic continuity while forcing the model to reconstruct the global semantic causal order.
>
> To quantitatively verify the effectiveness of DMSP, we compare our method with two variants: one baseline without sequence shuffling, and the other one replacing DMSP with the video shuffling method [2].  As shown in the table below, the video shuffling method [2] significantly underperforms our DMSP and even falls behind the baseline. It indicates that preserving local continuity is critical for learning motion representations.
>
> || HumanML3D ||| KIT-ML|||
> |-| - | - | - | - | - | - |
> || R@1| R@2 | R@3 | R@1| R@2| R@3 |
> | WaMo w/o shuffling| 12.24| 15.87| 22.46| 16.99| 23.13| 31.08|
> | WaMo w/ video shuffling [2] | 11.56| 15.42| 21.55| 15.03| 22.59| 28.89|
> | **WaMo w/ DMSP (Ours)**| **14.02** | **17.58** | **25.51** | **18.31** | **24.82** | **34.46** |
>
> > **Weakness 5 - Use of MoMask for evaluation**
>
> Thank you for your comment. The underlying representation learning in generation frameworks has demonstrated effectiveness for retrieval tasks [3, 4]. The adaptation has been proven effective in previous work [5].
>
> In particular, T2M [4] is also originally designed for text-to-motion generation and is adapted for retrieval [5]. It adds an additional GRU-based motion encoder to encode motion features extracted by T2M's VAE, then aligns motion features with text features using contrastive loss.
>
> Since T2M and MoMask [6] share a similar architecture, we strictly followed this approach to adapt MoMask for retrieval. The text features are extracted using CLIP. The motion sequence is first tokenized by MoMask's pretrained VQ-VAE, followed by an additional GRU-based motion encoder to obtain motion features. Next, the text-motion features are aligned using contrastive loss.
>
> ---
>
> References
>
> [1] Exploring Vision Transformers for 3D Human Motion-Language Models with Motion Patches, CVPR
>
> [2] Unsupervised Representation Learning by Sorting Sequences, CVPR
>
> [3] Temos: Generating diverse human motions from textual descriptions, ECCV
>
> [4] Generating diverse and natural 3d human motions from text, CVPR
>
> [5] TMR: Text-to-Motion Retrieval Using Contrastive 3D Human Motion Synthesis, CVPR
>
> [6] MoMask: Generative Masked Modeling of 3D Human Motions, CVPR

---

> ### Author Response · Authors · 2025-11-22
> **Response to Reviewer 1jTc (Part 3 of 4)**
>
> > **Weakness 6 - Low computational overhead of WaMo**
>
> We report the computational overhead and Text-to-Motion retrieval accuracy of the previous SOTA method MoPa, our base model without proposed modules (i.e., wavelet transforms and Disordered Motion Sequence Prediction), and our method on the HumanML3D dataset. The results are shown in the table below. The training process is equipped with early stopping.
>
> Compared to the base model, although our method leads to increased parameter size, memory usage, and training/inference time, it also significantly improves the retrieval accuracy. Compared to MoPa, we achieve better retrieval accuracy with faster inference speed and fewer parameters.
>
> In particular, although our method increases training time in one epoch, the total training time only slightly increases. This is due to our method converging using fewer epochs, indicating better learning efficiency.
>
> ||Parameters|Memory (Train)|Memory (Inference)|Train (Batch)|Train (Total)|Inference|R@1|R@2|R@3|
> |-|-|-|-|-|-|-|-|-|-|
> |MoPa|152.59M|19450M|1212 MB|99.61s|8019.34s|11.27s|10.80|14.98|20.00|
> |Base Model|**80.29M**|**18228M**|**672 MB**|**90.26s**|**6475.19s**| **7.21s**|9.14|13.65|18.72|
> |**Base Model + Ours (WaMo)**|84.34M|21258M|812 MB|156.64s|6749.12s|10.46s| **14.02**|**17.58**|**25.51**|
>
> > **Weakness 7 - Robustness and generalization ability of WaMo on larger and more diverse datasets**
>
> We further validate our method on the Motion-X++ dataset [1]. Compared to HumanML3D and KIT-ML, it is a larger dataset, containing longer motion sequences with longer and noisier text annotations (generated by LLMs) in more diverse scenarios.
>
> We report results of the previous SOTA method MoPa, our base model, and our full setup in the table below:
>
> ||Text-to-Motion|||Motion-to-Text|||
> |-|-|-|-|-|-|-|
> ||R@1|R@2|R@3|R@1|R@2|R@3|
> |MoPa|15.65|22.64|27.59|15.63|23.08|27.82|
> |Base Model|19.34|27.54|33.14|20.58|29.09|34.66|
> |**Base Model + Ours**|**24.11**|**33.11**|**39.20**|**24.19**| **34.84**|**40.85**|
>
> In particular, our method significantly outperforms MoPa. This is due to that MoPa pools joints into five body parts (torso, arms, legs), leading to significant information loss in dense skeletons (i.e., Motion-X++ with 52 joints compared to HumanML3D with 22 joints). In contrast, our method applies wavelet decomposition to the trajectory of every individual joint, preserving the fine-grained motion details and leading to better retrieval accuracy.
>
> ---
>
> References
>
> [1] Motion-X++: A Large-Scale Multimodal 3D Whole-body Human Motion Dataset, arXiv

---

> ### Author Response · Authors · 2025-11-22
> **Response to Reviewer 1jTc (Part 4 of 4)**
>
> > **Weakness 8 - Reproduced LaMP results**
>
> We have double-checked our implementation and ensured the results are correct. We reproduce LaMP [1] using the standard evaluation protocol. The relatively low performance compared to the original paper is due to the huge difference in the evaluation settings on the test batch scale.
>
> Analysis of the original LaMP paper and its reported TMR baselines suggests that the results were obtained using the "Small Batches" evaluation protocol. It performs retrieval within a small subset of **only 32 motion-text pairs**, which is a significantly easier setting compared to the standard protocol that involves retrieval across the **entire test set**. Therefore, a substantial performance drop when evaluating under the more challenging standard protocol is reasonable, which is consistent with other methods.
>
> Since the evaluation protocol is different, it is reasonable that LaMP might outperform baseline methods under "Small Batches", but underperform them under the standard protocol. Similarly, on the HumanML3D and KIT-ML datasets, T2M achieves higher Rsum scores than TEMOS under the "Small Batches" protocol (HumanML3D: 781.39 > 623.79, KIT-ML: 722.49 > 652.26). However, under the standard protocol, T2M underperforms TEMOS (HumanML3D: 63.57 < 72.58, KIT-ML: 129.39 < 207.84).
>
> > **Weakness 9 - Robustness and effectiveness of the Haar wavelet**
>
> The effectiveness of the Haar wavelet comes from its unique filter characteristics, which are suitable for analyzing human motions.
>
> The Haar wavelet has the shortest support length. This allows it to precisely capture short and abrupt motions (e.g., sudden acceleration of a limb).
>
> In contrast, higher-order Daubechies wavelets (e.g., db4, db8) use longer and smoother filters, which are more effective for continuous and smooth signals.  It tends to smooth the high-frequency local details, which are essential for fine-grained text-motion retrieval.
>
> Besides, the Haar wavelet achieves the best performance across both the HumanML3D (diverse daily actions) and KIT-ML (locomotion motions) datasets. The results are shown in the table below. This cross-dataset consistency demonstrates the robustness and generalization ability, which is not due to the dataset-specific bias.
>
> ||HumanML3D|||KIT-ML|||
> |-|-|-|-|-|-|-|
> |Wavelet|R@1|R@2|R@3|R@1|R@2|R@3|
> |db1|**14.02**|**17.58**|**25.51**|**18.31**|**24.82**|**34.46**|
> |db2|13.84|17.32|24.61|18.11|23.11|33.13|
> |db4|13.48|16.96|24.11|17.95|23.14|31.81|
> |db8|13.87|17.39|24.53|18.00|23.98|30.24|
> |db12|13.41|16.72|23.94|17.47|23.83|30.30|
> |bior3.1|13.76|16.80|24.28|18.10|24.10|32.41|
>
> ---
>
> References
>
> [1] Lamp: Language-motion pretraining for motion generation, retrieval, and captioning, ICLR

---

> > ### Comment · Reviewer_1jTc · 2025-11-26
> >
> > Thank you for your reply, but I still have some questions.
> >
> > 1. The procedure for using MoMask in retrieval requires a more detailed explanation. I still do not understand how the method is actually carried out. Does your approach require additional training? If so, the training process needs to be clearly described. In addition, why does your own training pipeline use BERT, while MoMask adopts CLIP? The rationale behind this inconsistency should be clarified.
> >
> > 2. Although the authors claim that Disordered Motion Sequence Prediction is specifically designed for the unique characteristics of motion data, I remain unconvinced. It appears to be a straightforward adaptation of a commonly used trick in image and video tasks, and I do not see a truly novel contribution here. DMSP seems to function as a plug-and-play module; if the authors believe it is a general-purpose design, please incorporate it into other motion-related methods to evaluate its effectiveness. It would even be informative to test it on other video tasks.
> >
> > 3. Decoupling motion in the frequency domain is a very common practice. The other reviewers and I cited numerous examples from other papers. In this paper, neither the use of frequency domain signals nor the stated motivation appears to be innovative. I do not find this aspect to constitute a meaningful contribution.
> >
> > 4. In the ablation study (Table 3), the performance on KIT-ML remains almost state-of-the-art even when all modules are disabled. How exactly is your baseline defined?
> >
> > 5. In Figure 1, I do not observe any essential differences among the three scales. For example, the left-arm trajectories appear flat across all three mid-scale segments, and the right-arm trajectories show large fluctuations at later stages for all scales. Thus, the motivation does not seem well justified. Furthermore, your method uses only high-frequency and low-frequency components, while the motivation in Figure 1 relies on three scales. This mismatch raises additional concerns about whether the motivation truly aligns with the method.

---

> > > ### Author Response · Authors · 2025-11-28
> > > **Response to Follow-up Questions (Part 1 of 3)**
> > >
> > > We sincerely appreciate your valuable time and thoughtful feedback on our response. Below we further provide detailed responses to the follow-up questions, which we hope could address your concerns:
> > >
> > > > **Discussion-Q1: Details about MoMask for retrieval**
> > >
> > > Thank you for your question. Following the adaptation method of T2M [1], we train the motion encoder and text encoder using contrastive loss [2] to align motion and text features. The loss is defined as: $ \mathcal{L}_{nce}=-\frac{1}{B} \sum\_{i}\left(\log \frac{\exp (S\_{i i} / \tau)}{\sum\_{j} \exp (S\_{i j} / \tau)}+\log \frac{\exp (S\_{i i} / \tau)}{\sum\_{j} \exp (S\_{j i} / \tau)}\right)$, where $B$ is the size of the mini-batch, $S\_{i j}=cos(t\_i, m\_j)$ is the cosine similarity of the $i$-th text and $j$-th motion features within the mini-batch, and $\tau$ is the temperature hyperparameter. The model is trained for 100 epochs with a batch size of 128. It is optimized using the Adam optimizer with a learning rate of 1e-4. The hidden size of the GRU-based motion encoder is set to 1024. $\tau$ is set to 0.07.
> > >
> > > We employ CLIP for MoMask, as MoMask itself originally uses CLIP for text feature extraction. Following prior works on text-motion retrieval [3, 4], our method adopts DistilBERT. To eliminate this inconsistency, we further adapt MoMask using DistilBERT. As shown in the table below, adopting CLIP or DistilBERT for MoMask achieves comparable results.
> > >
> > > || HumanML3D ||| KIT-ML |||
> > > | - | - | - | - | - | - | - |
> > > || R@1| R@2| R@3| R@1| R@2| R@3|
> > > | MoMask  w/ CLIP| 2.00| 3.58 | 5.03| 3.69| 7.81 | 12.22 |
> > > | MoMask w/ DistilBERT | 1.86 | 3.56 | 4.74 | 3.12 | 7.14 | 12.37 |
> > >
> > > > **Discussion-Q2: Novelty and generalization ability of Disordered Motion Sequence Prediction (DMSP)**
> > >
> > > We thank the reviewer for the insightful suggestion. We agree that DMSP can serve as a general-purpose plug-and-play module for motion representation learning.
> > >
> > > **Generalization to other methods.** To demonstrate the effectiveness of DMSP, following your suggestion, we first integrated it into two additional methods: T2M (GRU-based) [1] and TMR (Transformer-based) [3]. As shown in the table below, DMSP consistently achieves performance improvements across different model architectures.
> > >
> > > || HumanML3D ||| KIT-ML|||
> > > | - | - | - | - | - | - | - |
> > > || R@1| R@2| R@3| R@1| R@2| R@3|
> > > | T2M| 1.80| 3.42| 4.79| 3.37| 6.99| 10.84|
> > > | **T2M w/ DMSP** | **3.45**| **5.12** | **7.31** | **5.97** | **9.46**| **14.23** |
> > > | TMR| 5.68| 10.59| 14.04| 7.23| 13.98| 20.36|
> > > | **TMR w/ DMSP** |**8.19**|**12.75**|**16.87**| **9.64** | **16.78** | **23.47** |
> > >
> > > **Generalization to other motion-related tasks.** To further demonstrate the generalization ability of DMSP in other motion-related tasks, following MoPa [4], we further apply our method on two additional tasks: Zero-shot Motion Classification and Human Interaction Recognition.
> > >
> > > On Zero-shot Motion Classification, we adopt the BABEL 60-class benchmark [5], which contains 10892 sequences, and 20% of them are used as the test set. The motions are processed following the same procedure as HumanML3D [1]. We directly apply our model trained on HumanML3D to the test set. The action labels in BABEL are represented as “A person {action}” for classification. We then calculate the cosine similarity between a given motion and all 60 action labels. The action label with the highest similarity is taken as the final classification category. The Top-1 and Top-5 accuracy are shown in the table below, where DMSP leads to performance improvements.
> > >
> > > || Top-1| Top-5|
> > > | - | - | - |
> > > | TMR| 30.13| 41.52|
> > > | MoPa| 41.33| 68.97|
> > > | Ours w/o DMSP | 41.95| 69.29|
> > > | **Ours**| **43.19** | **70.21** |
> > >
> > > ---
> > >
> > > References
> > >
> > > [1] Generating diverse and natural 3d human motions from text, CVPR
> > >
> > > [2] Representation learning with contrastive predictive coding, arXiv
> > >
> > > [3] Tmr: Text-to-motion retrieval using contrastive 3d human motion synthesis, ICCV
> > >
> > > [4] Exploring vision transformers for 3d human motion-language models with motion patches, CVPR
> > >
> > > [5] Babel: Bodies, action and behavior with English labels, CVPR

---

> ### Author Response · Authors · 2025-11-28
> **Response to Follow-up Questions (Part 2 of 3)**
>
> >**Discussion-Q2  (continue)**
>
> On Human Interaction Recognition, we use the InterHuman Dataset [1] to demonstrate that our method can also be applied to multi-person motion recognition, beyond single-person motion recognition. InterHuman consists of diverse interactions between two individuals, which are split into 6222 samples for training and 1557 for testing. Following MoPa, we adopt a shared motion encoder for each individual’s motion. The motion features are then concatenated, followed by a projection layer. The results are shown in the table below, where DMSP leads to performance improvements in all metrics.
>
> ||Text-to-Motion|||Motion-to-Text|||
> -|-|-|-|-|-|-
> ||R@1|R@5|R@10|R@1|R@5|R@10|
> |TMR|5.38|15.64|24.40|5.13|15.26|25.65|
> |MoPa|9.51|21.27|32.41|8.26|22.65| 32.66|
> Ours w/o DMSP |10.86|24.09|35.68|10.56|24.53| 35.51
> **Ours**|**13.08**|**25.94**|**37.13**|**12.64**|**25.97**|**36.16**
>
> > **Discussion-Q3: Novelty of the use of frequency domain signals**
>
> Thank you for your insightful feedback. We agree that frequency transforms have been explored in prior single-modal motion-related works. While the quantitative experiments in our previous response have demonstrated the advantages of our proposed method over the related works mentioned by the reviewer, we would like to further clarify the unique novelty of our approach. Below, we emphasize the specific challenges of cross-modal text-motion retrieval that distinguish it from the mentioned single-modal motion generation/prediction tasks, and how our wavelet-based method addresses these challenges.
>
> Unlike single-modal motion generation/prediction (WaveAR [2], MotionWavelet [3], etc.), the core challenge of cross-modal text-motion retrieval lies in **bidirectional semantic alignment between text and motion**. Textual descriptions often contain fine-grained, discriminative details (e.g., "wave left hand quickly" vs. "wave right hand slowly") that must be precisely aligned with motion dynamics. Generation tasks mainly focus on motion synthesis and usually discard high-frequency components (as in HumanMAC [4] and LearnTrajDep [5]), but text-motion alignment requires preserving and exploiting these high-frequency components (e.g., abrupt limb movements) as they directly correspond to important textual semantics. The need for fine-grained alignment is unique to text-motion retrieval and not addressed in prior frequency-based motion works.
>
> Moreover, our Intra- and Inter-Frequency Attention module explicitly models the semantic dependencies between frequency components (e.g., how a high-frequency "hand wave" relates to the low-frequency "walking" context). Interpreting text that combines global and local motion semantics is an essential ability for text-motion retrieval. Prior wavelet-based works (WaveAR, MotionWavelet) do not model such frequency-semantic correlations, as they focus on single-modal motion prediction rather than cross-modal text-motion retrieval.
>
> > **Discussion-Q4: Implementation details of the base model**
>
> Thank you for the comment. In the base model, we do not apply wavelet transforms to motions and disable the reconstruction module and the Disordered Motion Sequence Prediction module. The text encoder remains unchanged. The primary difference between the base model and our full setup lies in the motion encoder. Below, we provide more implementation details of the motion encoder of the base model, which is stated in Line 403 of the paper.
>
> Given a motion sequence with $T$ frames and $J$ joints in $xyz$ coordinates, we denote it as ${M} \in \mathbb{R}^{T \times J \times 3}$. Then, $M$ is flattened, which is denoted as $\bar{M} \in \mathbb{R}^{T \times 3J}$. Next, we apply linear projection and transformer blocks to obtain the motion features $\hat{M}\in \mathbb{R}^{T \times D}$, where $D$ is the latent dimension. Finally, we employ additive attention pooling [6] on $\hat{M}$ to obtain the aggregated motion embedding $m \in \mathbb{R}^{D}$.
>
> On KIT-ML, even our base model achieves comparable performance to the previous SOTA method MoPa. This is because our base model preserves the fine-grained information of **every joint**. In contrast, MoPa pools joints into **five body parts** (torso, arms, legs) to simplify the skeletons, leading to information loss for specific joints. Moreover, adopting wavelet transforms further leverages the fine-grained information of every joint, leading to substantial performance improvements.
>
> ---
>
> References
>
> [1] Intergen: Diffusion-based multi-human motion generation under complex interactions, IJCV
>
> [2] WaveAR: Wavelet-Aware Continuous Autoregressive Diffusion for Accurate Human Motion Prediction, NeurIPS
>
> [3] MotionWavelet: Human Motion Prediction via Wavelet Manifold Learning, arXiv
>
> [4] HumanMAC: Masked Motion Completion for Human Motion Prediction, ICCV
>
> [5] Learning Trajectory Dependencies for Human Motion Prediction, ICCV
>
> [6] Neural machine translation by jointly learning to align and translate, ICLR

---

> ### Author Response · Authors · 2025-11-28
> **Response to Follow-up Questions (Part 3 of 3)**
>
> > **Discussion-Q5: Clarification of Figure 1**
>
> Thank you very much for the valuable comments and careful observation, which have helped us clarify important details about our work.
>
> **Clarification on the waveform of the left arm.** As described in the text query ("The person pulls something towards him with his right hand, picks something up with his left hand, wipes it off using right hand and puts it down in his left hand"), the left arm has relatively fewer movements compared to the right arm, exhibiting only two distinct actions: "picks something up with his left hand" and "puts it down in his left hand". These specific actions are reflected in the two red fluctuations in scale 0 and scale 1. The left arm does not move between 50–100 frames, showing a flat trajectory. Notably, the "picks something up with his left hand" action is not shown in scale 2, while the "puts it down in his left hand" action is clearly highlighted in scale 2. This is due to scale 2 captures higher-frequency actions, so the low-frequency "pick" action is filtered.
>
> **Clarification on the waveform of the right arm.** For the right arm at later stages, while all scales exhibit large fluctuations, scale 0 preserves the overall structure of the original motion sequence. Although both scale 1 and scale 2 capture high-frequency signals, scale 2 remains flat during earlier stages and only shows fluctuations at the later stage. This observation aligns with the described human motion: when the right hand passes an object to the left hand, it makes subtle, fine-grained movements to achieve precise positioning before release. It shows that scale 2 filters out low- and mid-high-frequency information from earlier stages, highlighting only the higher-frequency signals of the two-handed interaction. This is consistent with our motivation to capture different frequency components, each with discriminative semantics that correspond to the text query.
>
> **Clarification on the use of multi-scale features.** We would like to clarify the confusion about the mismatch between the three-scale frequency components in Figure 1 and our use of high/low-frequency components. In Figure 1, scale 0 corresponds to the low-frequency component, and **both scale 1 and scale 2 represent high-frequency components**. This is because wavelet transforms lead to one low-frequency component and several high-frequency components. **All of the decomposed frequency components are used in our method.**
>
> We further demonstrate the effectiveness of multi-scale decomposition. The retrieval accuracy across different wavelet decomposition levels $S$ is shown in the table below.
>
> || HumanML3D ||| KIT-ML|||
> | - | - | - | - | - | - | - |
> | $S$| R@1| R@2| R@3| R@1| R@2| R@3|
> | 1| 11.93| 15.75| 22.21| 15.66| 22.85| 30.26|
> | 2| 12.81| 16.39| 23.16| 16.99| 23.73| 30.96|
> | **3** | **14.02** | **17.58** | **25.51** | **18.31** | **24.82** | **34.46** |
> | 4| 13.84| 17.20| 24.52| 18.19| 24.06| 31.33|
>
> When the decomposition level increases from $S=1$ to $S=3$, all metrics improve significantly. It demonstrates that deeper decomposition captures richer temporal hierarchies. However, at $S=4$, there is a slight drop in all metrics. It suggests that excessive decomposition may introduce noise and motion-irrelevant features, limiting precise text-motion matching. These quantitative results further justify our motivation for multi-scale motion decomposition.
>
> ---
>
> Thank you again for your expertise and time. We are fully open to any additional suggestions that could further improve our study.

---

### Official Review · Reviewer_YAar · 2025-10-31

**Soundness:** 3
**Presentation:** 3
**Contribution:** 3
**Rating:** 4
**Confidence:** 5

**Summary:**

This paper introduces WaMo, a method for fine-grained text-to-motion retrieval. To better capture the complex temporal dynamics of human motion, WaMo decomposes joint trajectories into multiple frequency bands using wavelets, reconstructs them to preserve spatial-temporal structure, and employs a disordered sequence prediction task to encourage understanding of temporal order. By combining multi-frequency motion analysis with temporal reasoning, the model aims to align motion sequences more accurately with textual descriptions. Experiments on HumanML3D and KIT-ML demonstrate consistent improvements over prior approaches.

**Strengths:**

•	Introduces wavelet-based multi-frequency analysis for fine-grained motion modeling.

•	Focuses on part-specific and temporal dynamics, improving text-motion alignment.

•	Achieves clear empirical gains on standard benchmarks.

•	Provides a coherent, modular pipeline (TWD, TWR, DMSP) from decomposition to retrieval embedding.

**Weaknesses:**

1. **Conceptual novelty and task-specific motivation are unclear.**
WaMo combines multi-frequency decomposition and disordered motion sequence prediction (DMSP), but each component has clear precedents. Frequency-based representations have been used in motion-related tasks (e.g., WaveletMotion, WaveAR, HumanMAC), local/part-level modeling is already incorporated in methods like MoPa（MotionPatch，CVPR2024）, and shuffling/reordering sequences resembles common self-supervised tasks in video and image domains. It remains unclear what is genuinely novel for text-to-motion retrieval. The authors should clarify: which aspects of multi-frequency decomposition are retrieval-specific, and whether DMSP includes any motion-specific adaptations.

2.	**Ambiguity in evaluation using MoMask.**
Since MoMask was originally designed for text-to-motion generation, it is unclear how it is adapted for retrieval. More details on the evaluation procedure are needed to ensure fairness and reproducibility.

3.	**Computational overhead not discussed.**
Frequency decomposition adds processing cost, but the paper does not report memory usage, latency, or parameter increase. Including such analysis would help assess scalability and practical feasibility.

4.	**Limited dataset diversity.**
Experiments are confined to HumanML3D and KIT-ML, which are relatively small and clean. Evaluation on larger or more diverse datasets (e.g., Motion-X) would better demonstrate robustness and generalization.

5.	**Unexpectedly low LaMP results.**
Reported LaMP performance is lower than in the original paper, where it slightly outperformed TMR baselines. The evaluation protocol or implementation should be double-checked to ensure comparability.

6. **Sensitivity to wavelet basis choice.**
Table 10 shows performance is strongly dependent on the wavelet initialization (e.g., db1 performs best). The authors should discuss why Haar (db1) is particularly effective and whether this reflects dataset-specific bias or intrinsic alignment with certain motion patterns.

**Questions:**

None

---

> ### Author Response · Authors · 2025-11-22
> **Response to Reviewer YAar (Part 1 of 3)**
>
> Thank you for your valuable time and thoughtful feedback on our manuscript, which has greatly helped us improve it. We address each of your points below.
>
> > **Weakness 1 - Conceptual novelty and task-specific motivation**
>
> Thank you for your comment. We clarify the novelty of our method in the responses below. The discussions have been incorporated into the revised paper.
>
> > **Weakness 1.1 - Frequency-based representations**
>
> While frequency decomposition has been used in single-modal motion prediction and generation [1-4], we are the first to introduce multi-frequency analysis specifically for cross-modal Text-Motion Retrieval (TMR). The novelty of WaMo for TMR lies in semantic alignment. Both text queries and motions inherently contain multi-granular information, ranging from global trends (low-frequency) to fine-grained details (high-frequency). WaMo employs wavelet decomposition not for smoothing or prediction, but to explicitly decompose these motion components, enabling precise alignment with the corresponding textual semantics.
>
> Besides, HumanMAC [1] and LearnTrajDep [2] use the Discrete Cosine Transform (DCT) to capture global dynamics for motion generation and prediction, but they discard high-frequency components to smooth human motion. However, in TMR, high-frequency details are usually semantically significant features (e.g., short and abrupt motion). Instead of smoothing motions, WaMo employs wavelets to extract these discriminative features to match specific semantics in the text query.
>
> WaveAR [3] and MotionWavelet [4] adopt wavelets to model non-stationary dynamics for motion prediction. However, they do not distinguish the fine-grained semantics of low- and high-frequency components. In TMR, the text queries describe motions at varying granularities, from global trends (e.g., walking) related to low-frequency components, to short and abrupt actions (e.g., sudden acceleration of a limb) related to high-frequency components. Therefore, we further capture both frequency-specific characteristics and their inter-dependencies through the Intra- and Inter-Frequency Attention module, facilitating fine-grained alignment with textual semantics.
>
> To quantitatively verify the effectiveness of our design, we compare WaMo with two variants: one replacing wavelets with DCT (corresponding to HumanMAC and LearnTrajDep), and one removing the Intra-Frequency Attention (corresponding to WaveAR and MotionWavelet). As shown in the table below, both modifications lead to clear performance drops, indicating the effectiveness of preserving high-frequency information and modeling intra- and inter-frequency semantics.
>
> || HumanML3D ||| KIT-ML|||
> | - | - | - | - | - | - | - |
> || R@1| R@2 | R@3 | R@1| R@2| R@3 |
> | WaMo w/ DCT| 12.69| 16.10| 22.84| 16.17| 23.96| 31.15|
> | WaMo w/o Intra-Frequency Attention | 12.82| 15.74| 23.36| 17.11| 22.71| 30.12|
> | **WaMo (Ours)**| **14.02** | **17.58** | **25.51** | **18.31** | **24.82** | **34.46** |
>
> > **Weakness 1.2 - Local/part-level modeling**
>
> We thank the reviewer for this comment. We would like to clarify that MoPa [5] uses a **coarse-grained** method to model spatiotemporal structure, limiting its effectiveness on dense skeletons. In contrast, WaMo captures **fine-grained** motion dynamics, leading to better preservation of complex kinematic details.
>
> MoPa pools joints into **five body parts** (torso, arms, legs) to simplify the skeletons. In contrast, WaMo applies wavelet decomposition to the trajectory of **every** **individual joint**. The temporal dynamics of every joint are decomposed into multi-scale frequency bands to capture fine-grained motion characteristics, which coarse-grained part-level pooling may overlook.
>
> Besides, MoPa's joint pooling causes information loss for specific joints, while WaMo preserves the fine-grained information of every joint. It is particularly critical for dense skeletons (e.g., Motion-X++ [6] with 52 joints compared to HumanML3D with 22 joints). On the Motion-X++ dataset, WaMo significantly outperforms MoPa, indicating the effectiveness of WaMo in capturing the complex dynamics of dense skeletons. Please refer to the response to Weakness 4 for results on Motion-X++.
>
> ---
>
> References
>
> [1] HumanMAC: Masked Motion Completion for Human Motion Prediction, ICCV
>
> [2] Learning Trajectory Dependencies for Human Motion Prediction, ICCV
>
> [3] WaveAR: Wavelet-Aware Continuous Autoregressive Diffusion for Accurate Human Motion Prediction, NeurIPS
>
> [4] MotionWavelet: Human Motion Prediction via Wavelet Manifold Learning, arXiv
>
> [5] Exploring Vision Transformers for 3D Human Motion-Language Models with Motion Patches, CVPR
>
> [6] Motion-X++: A Large-Scale Multimodal 3D Whole-body Human Motion Dataset, arXiv

---

> ### Author Response · Authors · 2025-11-22
> **Response to Reviewer YAar (Part 2 of 3)**
>
> > **Weakness 1.3 - Shuffling/Reordering sequences**
>
> We appreciate the reviewer’s feedback. While sequence reordering is common in 2D vision, its direct application to 3D human motion leads to performance degradations due to the fundamental difference in visual modality.
>
> Unlike 2D images/videos that contain dense static visual textures (e.g., background, object appearance) to provide context, 3D human motion is sparse and represented by kinematic dynamics (e.g., velocity and acceleration).
>
> Frame-level shuffling (as used in video tasks [1]) rearranges every frame across the entire sequence. Directly applying it to motion completely disrupts these local kinematic dependencies. It results in discontinuous, physically impossible actions rather than meaningful motion segments, confusing the encoder.
>
> In contrast, DMSP is specifically designed for the unique nature of motion. By dividing sequences into temporally coherent groups and only shuffling a small group of frames, DMSP preserves the local kinematic continuity while forcing the model to reconstruct the global semantic causal order.
>
> To quantitatively verify the effectiveness of DMSP, we compare our method with two variants: one baseline without sequence shuffling, and the other one replacing DMSP with the video shuffling method [1].  As shown in the table below, the video shuffling method [1] significantly underperforms our DMSP and even falls behind the baseline. It indicates that preserving local continuity is critical for learning motion representations.
>
> || HumanML3D ||| KIT-ML|||
> |-| - | - | - | - | - | - |
> || R@1| R@2 | R@3 | R@1| R@2| R@3 |
> | WaMo w/o shuffling| 12.24| 15.87| 22.46| 16.99| 23.13| 31.08|
> | WaMo w/ video shuffling [1] | 11.56| 15.42| 21.55| 15.03| 22.59| 28.89|
> | **WaMo w/ DMSP (Ours)**| **14.02** | **17.58** | **25.51** | **18.31** | **24.82** | **34.46** |
>
> > **Weakness 2 - Use of MoMask for evaluation**
>
> Thank you for your comment. The underlying representation learning in generation frameworks has demonstrated effectiveness for retrieval tasks [2, 3]. The adaptation has been proven effective in previous work [4].
>
> In particular, T2M [3] is also originally designed for text-to-motion generation and is adapted for retrieval [4]. It adds an additional GRU-based motion encoder to encode motion features extracted by T2M's VAE, then aligns motion features with text features using contrastive loss.
>
> Since T2M and MoMask [5] share a similar architecture, we strictly followed this approach to adapt MoMask for retrieval. The text features are extracted using CLIP. The motion sequence is first tokenized by MoMask's pretrained VQ-VAE, followed by an additional GRU-based motion encoder to obtain motion features. Next, the text-motion features are aligned using contrastive loss.
>
>
> > **Weakness 3 - Low computational overhead of WaMo**
>
> We report the computational overhead and Text-to-Motion retrieval accuracy of the previous SOTA method MoPa [6], our base model without proposed modules (i.e., wavelet transforms and Disordered Motion Sequence Prediction), and our method on the HumanML3D dataset. The results are shown in the table below. The training process is equipped with early stopping.
>
> Compared to the base model, although our method leads to increased parameter size, memory usage, and training/inference time, it also significantly improves the retrieval accuracy. Compared to MoPa, we achieve better retrieval accuracy with faster inference speed and fewer parameters.
>
> In particular, although our method increases training time in one epoch, the total training time only slightly increases. This is due to our method converging using fewer epochs, indicating better learning efficiency.
>
> || Parameters | Memory (Train) | Memory (Inference) | Train (Batch) | Train (Total) | Inference | R@1| R@2| R@3|
> | - | - | - | - | - | - | - | - | - | - |
> | MoPa| 152.59M| 19450M| 1212 MB| 99.61s| 8019.34s| 11.27s| 10.80| 14.98| 20.00|
> | Base Model| **80.29M** | **18228M**| **672 MB**| **90.26s**| **6475.19s**  | **7.21s** | 9.14| 13.65| 18.72|
> | **Base Model + Ours (WaMo)** | 84.34M| 21258M| 812 MB| 156.64s| 6749.12s| 10.46s| **14.02** | **17.58** | **25.51** |
>
> ---
>
> References
>
> [1] Unsupervised Representation Learning by Sorting Sequences, CVPR
>
> [2] Temos: Generating diverse human motions from textual descriptions, ECCV
>
> [3] Generating diverse and natural 3d human motions from text, CVPR
>
> [4] TMR: Text-to-Motion Retrieval Using Contrastive 3D Human Motion Synthesis, CVPR
>
> [5] MoMask: Generative Masked Modeling of 3D Human Motions, CVPR
>
> [6] Exploring Vision Transformers for 3D Human Motion-Language Models with Motion Patches, CVPR

---

> ### Author Response · Authors · 2025-11-22
> **Response to Reviewer YAar (Part 3 of 3)**
>
> > **Weakness 4 - Robustness and generalization ability of WaMo on larger and more diverse datasets**
>
> We further validate our method on the Motion-X++ dataset [1]. Compared to HumanML3D and KIT-ML, it is a larger dataset, containing longer motion sequences with longer and noisier text annotations (generated by LLMs) in more diverse scenarios.
>
> We report results of the previous SOTA method MoPa, our base model, and our full setup in the table below:
>
> || Text-to-Motion ||| Motion-to-Text |||
> | - | - | - | - | - | - | - |
> || R@1| R@2| R@3| R@1| R@2| R@3|
> | MoPa| 15.65| 22.64| 27.59| 15.63| 23.08| 27.82|
> | Base Model| 19.34| 27.54| 33.14| 20.58| 29.09| 34.66|
> | **Base Model + Ours** | **24.11**| **33.11** | **39.20** | **24.19**| **34.84** | **40.85** |
>
> In particular, our method significantly outperforms MoPa. This is due to that MoPa pools joints into five body parts (torso, arms, legs), leading to significant information loss in dense skeletons (i.e., Motion-X++ with 52 joints compared to HumanML3D with 22 joints). In contrast, our method applies wavelet decomposition to the trajectory of every individual joint, preserving the fine-grained motion details and leading to better retrieval accuracy.
>
> > **Weakness 5 - Reproduced LaMP results**
>
> We have double-checked our implementation and ensured the results are correct. We reproduce LaMP [2] using the standard evaluation protocol. The relatively low performance compared to the original paper is due to the huge difference in the evaluation settings on the test batch scale.
>
> Analysis of the original LaMP paper and its reported TMR baselines suggests that the results were obtained using the "Small Batches" evaluation protocol. It performs retrieval within a small subset of **only 32 motion-text pairs**, which is a significantly easier setting compared to the standard protocol that involves retrieval across the **entire test set**. Therefore, a substantial performance drop when evaluating under the more challenging standard protocol is reasonable, which is consistent with other methods.
>
> Since the evaluation protocol is different, it is reasonable that LaMP might outperform baseline methods under "Small Batches", but underperform them under the standard protocol. Similarly, on the HumanML3D and KIT-ML datasets, T2M achieves higher Rsum scores than TEMOS under the "Small Batches" protocol (HumanML3D: 781.39 > 623.79, KIT-ML: 722.49 > 652.26). However, under the standard protocol, T2M underperforms TEMOS (HumanML3D: 63.57 < 72.58, KIT-ML: 129.39 < 207.84).
>
> > **Weakness 6 - Robustness and effectiveness of the Haar wavelet**
>
> The effectiveness of the Haar wavelet comes from its unique filter characteristics, which are suitable for analyzing human motions.
>
> The Haar wavelet has the shortest support length. This allows it to precisely capture short and abrupt motions (e.g., sudden acceleration of a limb).
>
> In contrast, higher-order Daubechies wavelets (e.g., db4, db8) use longer and smoother filters, which are more effective for continuous and smooth signals.  It tends to smooth the high-frequency local details, which are essential for fine-grained text-motion retrieval.
>
> Besides, the Haar wavelet achieves the best performance across both the HumanML3D (diverse daily actions) and KIT-ML (locomotion motions) datasets. The results are shown in the table below. This cross-dataset consistency demonstrates the robustness and generalization ability, which is not due to the dataset-specific bias.
>
> || HumanML3D ||| KIT-ML|||
> | - | - | - | - | - | - | - |
> | Wavelet | R@1| R@2 | R@3 | R@1| R@2| R@3 |
> | db1| **14.02** | **17.58** | **25.51** | **18.31** | **24.82** | **34.46** |
> | db2| 13.84| 17.32| 24.61| 18.11| 23.11| 33.13|
> | db4| 13.48| 16.96| 24.11| 17.95| 23.14| 31.81|
> | db8| 13.87| 17.39| 24.53| 18.00| 23.98| 30.24|
> | db12| 13.41| 16.72| 23.94| 17.47| 23.83| 30.30|
> | bior3.1 | 13.76| 16.80| 24.28| 18.10| 24.10| 32.41|
>
> ---
>
> References
>
> [1] Motion-X++: A Large-Scale Multimodal 3D Whole-body Human Motion Dataset, arXiv
>
> [2] Lamp: Language-motion pretraining for motion generation, retrieval, and captioning, ICLR

---

> > ### Comment · Reviewer_YAar · 2025-11-27
> >
> > 1. While the authors correctly identify the shortest support of the Haar wavelet as a theoretical advantage for capturing short, abrupt motions, this temporal strength is counterbalanced by a significant weakness in the frequency domain. The Haar wavelet suffers from slow spectral decay, leading to very poor frequency resolution. This means it is ineffective at separating signal components with closely spaced frequencies. In practice, the Haar wavelet behaves more like a differential operator than a true bandpass filter. Consequently, a critical question arises: can a wavelet with such inadequate frequency characterization truly satisfy the paper's motivation for analyzing human motion? Human motion is often composed of complex, overlapping frequency components, such as the superposition of a primary limb movement with smaller, jittery micro-motions. The Haar wavelet's inability to discriminate between these components may oversimplify the analysis and obscure important dynamic details. Therefore, its suitability for the proposed task remains questionable.
> > 2. Both the use of DMSP and the frequency-domain analysis are well-established methods in the field. I find the method approach to lack significant novelty.
> > 3. I still have reservations regarding the applicability of the Momask method to the retrieval task.

---

> > > ### Author Response · Authors · 2025-11-28
> > > **Response to Follow-up Questions (Part 3 of 3)**
> > >
> > > > **Discussion-Q3:  Applicability of MoMask for retrieval**
> > >
> > > Thank you for the follow-up comment. We understand your reservation regarding the difference between generation and retrieval objectives. We would like to further clarify the rationale and empirical validity of including MoMask [1] as a baseline.
> > >
> > > While MoMask is designed for generation, its vector-quantized (VQ) latent space is highly expressive. Recent works on text-image retrieval [2] have shown that latent spaces of text-to-image generation models provide strong representations for cross-modal retrieval. After training with contrastive learning, they can serve as strong baselines for retrieval.
> > >
> > > Following this insight, our intention was to evaluate whether the discrete tokens learned by MoMask's VQ-VAE capture sufficient semantics for retrieval when adapted.
> > >
> > > As shown in Section 5.2, we compare with a wide range of methods, including both retrieval methods (e.g., TMR [3], MoPa [4]) and other adapted generation methods (e.g., T2M [5], TEMOS [6]). Including MoMask ensures a comprehensive comparison across different model architectures. In our experiments, the adapted MoMask achieved reasonable retrieval performance compared to other adapted generation methods.
> > >
> > > To address your concern about ambiguity, we have explicitly categorized baselines into "Text-motion retrieval models" and "Text-motion generation models" in Table 1 and Table 2 to avoid confusion.
> > >
> > > We hope this explanation alleviates your concerns regarding the applicability of MoMask in the retrieval task.
> > >
> > > ---
> > >
> > > References
> > >
> > > [1] MoMask: Generative Masked Modeling of 3D Human Motions, CVPR
> > >
> > > [2] Discffusion: Discriminative diffusion models as few-shot vision and language learners, TMLR
> > >
> > > [3] Tmr: Text-to-motion retrieval using contrastive 3d human motion synthesis, ICCV
> > >
> > > [4] Exploring vision transformers for 3d human motion-language models with motion patches, CVPR
> > >
> > > [5] Generating diverse and natural 3d human motions from text, CVPR
> > >
> > > [6] Temos: Generating diverse human motions from textual descriptions, ECCV
> > >
> > > ---
> > >
> > > Thank you again for your expertise and time. We are fully open to any additional suggestions that could further improve our study.

---

> ### Author Response · Authors · 2025-11-28
> **Response to Follow-up Questions (Part 1 of 3)**
>
> We sincerely appreciate your valuable time and thoughtful feedback on our response. Below we further provide detailed responses to the follow-up questions, which we hope could address your concerns:
>
> > **Discussion-Q1: Use of the Haar wavelet**
>
> We thank the reviewer for the insightful comment and acknowledgement of the key theoretical advantage of the Haar wavelet's shortest support for capturing short, abrupt motions. We agree with the theoretical analysis about the limitations of the Haar wavelet. Its slow spectral decay leads to limited frequency resolution.
>
> However, the fundamental trade-off in signal processing constrains that it is hard to simultaneously achieve optimal resolution in both time and frequency domains. For Text-Motion Retrieval, we argue that time-domain precision is more important than frequency-domain smoothness. Text descriptions of human motion typically focus on temporal events and transitions (e.g., "sudden stop," "kick," "turn quickly") rather than global spectral characteristics. These semantics are strictly time-localized. The Haar wavelet preserves temporal localization aligning with these textual descriptions.
>
> We also want to emphasize that human motion signals are non-stationary, containing many abrupt actions (e.g., sudden stops). Higher-order wavelets (e.g., db4) provide better frequency resolution but introduce ringing artifacts (Gibbs phenomenon) in the time domain. These oscillations can be mistakenly identified as actual motion jitter by the network, limiting the retrieval performance. In contrast, the Haar wavelet has the shortest support, which does not suffer from ringing artifacts. It preserves the accuracy of motion boundaries.
>
> Moreover, the wavelet filters in our method are learnable, not fixed. The learnable wavelet filters are initialized using the Haar wavelet. During training, it is optimized to capture specific, non-smooth motion dynamics that are semantically relevant to the text descriptions.
>
> To quantitatively verify the effectiveness of learnable wavelets, we compare our method with a variant using the fixed Haar wavelet. As shown in the table below, using learnable wavelets leads to better retrieval accuracy across all metrics on both datasets.
>
> ||HumanML3D|||KIT-ML|||
> |-| - | - | - | - | - | - |
> ||R@1|R@2 |R@3 | R@1| R@2 | R@3 |
> |WaMo w/ fixed wavelet| 12.56| 15.67| 22.64| 16.37| 23.29| 31.14|
> |**WaMo w/ learnable wavelet (Ours)**| **14.02**| **17.58**| **25.51**| **18.31** | **24.82** | **34.46** |
>
> > **Discussion-Q2: Novelty of our method**
>
> Thank you for your comment. We clarify the novelty of our method in the responses below.
>
>
> > **Discussion-Q2.1: Novelty of frequency-domain analysis**
>
> We agree that frequency transforms have been explored in prior single-modal motion-related works. While the quantitative experiments in our previous response have demonstrated the advantages of our proposed method over the related works mentioned by the reviewer, we would like to further clarify the unique novelty of our approach. Below, we emphasize the specific challenges of cross-modal text-motion retrieval that distinguish it from the mentioned single-modal motion generation/prediction tasks, and how our wavelet-based method addresses these challenges.
>
> Unlike single-modal motion generation/prediction (WaveAR [1], MotionWavelet [2], etc.), the core challenge of cross-modal text-motion retrieval lies in **bidirectional semantic alignment between text and motion**. Textual descriptions often contain fine-grained, discriminative details (e.g., "wave left hand quickly" vs. "wave right hand slowly") that must be precisely aligned with motion dynamics. Generation tasks mainly focus on motion synthesis and usually discard high-frequency components (as in HumanMAC [3] and LearnTrajDep [4]), but text-motion alignment requires preserving and exploiting these high-frequency components (e.g., abrupt limb movements) as they directly correspond to important textual semantics. The need for fine-grained alignment is unique to text-motion retrieval and not addressed in prior frequency-based motion works.
>
> Moreover, our Intra- and Inter-Frequency Attention module explicitly models the semantic dependencies between frequency components (e.g., how a high-frequency "hand wave" relates to the low-frequency "walking" context). Interpreting text that combines global and local motion semantics is an essential ability for text-motion retrieval. Prior wavelet-based works (WaveAR, MotionWavelet) do not model such frequency-semantic correlations, as they focus on single-modal motion prediction rather than cross-modal text-motion retrieval.
>
> ---
>
> References
>
> [1] WaveAR: Wavelet-Aware Continuous Autoregressive Diffusion for Accurate Human Motion Prediction, NeurIPS
>
> [2] MotionWavelet: Human Motion Prediction via Wavelet Manifold Learning, arXiv
>
> [3] HumanMAC: Masked Motion Completion for Human Motion Prediction, ICCV
>
> [4] Learning Trajectory Dependencies for Human Motion Prediction, ICCV

---

> ### Author Response · Authors · 2025-11-28
> **Response to Follow-up Questions (Part 2 of 3)**
>
> > **Discussion-Q2.2: Novelty and generalization ability of Disordered Motion Sequence Prediction (DMSP)**
>
> We would like to clarify that our DMSP distinguishes from methods in image and video tasks, and quantitative experiments in our previous response have demonstrated the advantages of our proposed method. In this response, we further demonstrate that DMSP can serve as a general-purpose plug-and-play module for motion representation learning.
>
> **Generalization to other methods.** To demonstrate the effectiveness of DMSP, we first integrated it into two additional methods: T2M (GRU-based) [1] and TMR (Transformer-based) [2]. As shown in the table below, DMSP consistently achieves performance improvements across different model architectures.
>
> || HumanML3D ||| KIT-ML|||
> | - | - | - | - | - | - | - |
> || R@1| R@2| R@3| R@1| R@2| R@3|
> | T2M| 1.80| 3.42| 4.79| 3.37| 6.99| 10.84|
> | **T2M w/ DMSP** | **3.45**  | **5.12**  | **7.31**  | **5.97** | **9.46**  | **14.23** |
> | TMR| 5.68| 10.59| 14.04| 7.23| 13.98| 20.36|
> | **TMR w/ DMSP** | **8.19**  | **12.75** | **16.87** | **9.64** | **16.78** | **23.47** |
>
> **Generalization to other motion-related tasks.** To further demonstrate the generalization ability of DMSP in other motion-related tasks, following MoPa [3], we further apply our method on two additional tasks: Zero-shot Motion Classification and Human Interaction Recognition.
>
> On Zero-shot Motion Classification, we adopt the BABEL 60-class benchmark [4], which contains 10892 sequences, and 20% of them are used as the test set. The motions are processed following the same procedure as HumanML3D [1]. We directly apply our model trained on HumanML3D to the test set. The action labels in BABEL are represented as “A person {action}” for classification. We then calculate the cosine similarity between a given motion and all 60 action labels. The action label with the highest similarity is taken as the final classification category. The Top-1 and Top-5 accuracy are shown in the table below, where DMSP leads to performance improvements.
>
> || Top-1| Top-5|
> | - | - | - |
> | TMR| 30.13| 41.52|
> | MoPa| 41.33| 68.97|
> | Ours w/o DMSP | 41.95| 69.29|
> | **Ours**| **43.19** | **70.21** |
>
> On Human Interaction Recognition, we use the InterHuman Dataset [5] to demonstrate that our method can also be applied to multi-person motion recognition, beyond single-person motion recognition. InterHuman consists of diverse interactions between two individuals, which are split into 6222 samples for training and 1557 for testing. Following MoPa, we adopt a shared motion encoder for each individual’s motion. The motion features are then concatenated, followed by a projection layer. The results are shown in the table below, where DMSP leads to performance improvements in all metrics.
>
> || Text-to-Motion ||| Motion-to-Text |||
> | - | - | - | - | - | - | - |
> || R@1| R@5| R@10| R@1| R@5| R@10|
> | TMR| 5.38| 15.64| 24.40| 5.13| 15.26| 25.65|
> | MoPa| 9.51| 21.27| 32.41| 8.26| 22.65| 32.66|
> | Ours w/o DMSP | 10.86| 24.09| 35.68| 10.56| 24.53| 35.51|
> | **Ours**| **13.08**| **25.94** | **37.13** | **12.64**| **25.97** | **36.16** |
>
> ---
>
> References
>
> [1] Generating diverse and natural 3d human motions from text, CVPR
>
> [2] Tmr: Text-to-motion retrieval using contrastive 3d human motion synthesis, ICCV
>
> [3] Exploring Vision Transformers for 3D Human Motion-Language Models with Motion Patches, CVPR
>
> [4] Babel: Bodies, action and behavior with English labels, CVPR
>
> [5] Intergen: Diffusion-based multi-human motion generation under complex interactions, IJCV

---

### Official Review · Reviewer_j2hX · 2025-10-31

**Soundness:** 4
**Presentation:** 4
**Contribution:** 3
**Rating:** 8
**Confidence:** 3

**Summary:**

The paper proposes a new wavelet-based multi-frequency feature extraction framework (thus a representation) that contributes to significantly advancing the SoTA on the Text-Motion Retrieval task.

The three modules proposed (Trajectory Wavelet Decomposition, Reconstruction, and Disordered Motion Sequence Prediction appear to significantly improve the learning of the temporal structures.

Results are impressive. The quality of the presentation is high.

The experimental settings and discussion are trustworthy and complete.

The appendix contains many details and is useful.

The only limitation is that the proposed approach is specific to the given task. There is no discussion regarding other possible applications of the proposed approach to different tasks and domains.

**Strengths:**

- impressive results
- trustable experimental settings
- comprehensive ablations
- qualitative evidence
- theoretical grounding

**Weaknesses:**

- the proposed approach is very specific for the given task (3D motion retrieval). This could limit the interest for the overall CLR community
- the contribution is rather architectural than conceptual

**Questions:**

Could you discuss how you expect the approach to perform on longer or noiser motion sequences (i.e., beyond the datasets tested that are the ones available in the literature)? Just you opinion...

Would it be possible to interpret the learned wavelet filters visually?

---

> ### Author Response · Authors · 2025-11-22
> **Response to Reviewer j2hX (Part 1 of 2)**
>
> Thank you for your valuable time and thoughtful feedback on our manuscript, which has greatly helped us improve it. We address each of your points below.
>
> > **Weakness 1 - Generalization ability beyond the given task (3D motion retrieval)**
>
> We would like to clarify that 3D motion retrieval is a fundamental task in multimodal text-motion understanding. To demonstrate the generalization ability of our method, following MoPa [1], we further apply our method on **two additional tasks**: Zero-shot Motion Classification and Human Interaction Recognition.
>
> On Zero-shot Motion Classification, we adopt the BABEL 60-class benchmark [2], which contains 10892 sequences, and 20% of them are used as the test set. The motions are processed following the same procedure as HumanML3D [3]. We directly apply our model trained on HumanML3D to the test set. The action labels in BABEL are represented as “A person {action}” for classification. We then calculate the cosine similarity between a given motion and all 60 action labels. The action label with the highest similarity is taken as the final classification category. The Top-1 and Top-5 accuracy are shown in the table below, where our method outperforms TMR [4] and MoPa.
>
> ||Top-1|Top-5|
> |-|-|-|
> |TMR|30.13|41.52|
> |MoPa|41.33|68.97|
> |**Ours**|**43.19**|**70.21**|
>
> On Human Interaction Recognition, we use the InterHuman Dataset [5] to demonstrate that our method can also be applied to multi-person motion recognition, beyond single-person motion recognition. InterHuman consists of diverse interactions between two individuals, which are split into 6222 samples for training and 1557 for testing. Following MoPa, we adopt a shared motion encoder for each individual’s motion. The motion features are then concatenated, followed by a projection layer. The results are shown in the table below, where our method outperforms TMR and MoPa.
>
> ||Text-to-Motion||  |Motion-to-Text|||
> |-|-|-|-|-|-|-|
> ||R@1|R@5|R@10|R@1|R@5|R@10|
> |TMR|5.38|15.64|24.40|5.13|15.26|25.65|
> |MoPa|9.51|21.27|32.41|8.26|22.65|32.66|
> |**Ours**|**13.08**|**25.94**|**37.13**|**12.64**|**25.97**|**36.16**|
>
> > **Weakness 2 - The conceptual contribution of WaMo**
>
> Thank you for your comment. While the architectural novelty is clear, we would like to further clarify our conceptual contribution.
>
> Firstly, we model motion as **multi-frequency components**. Prior works on 3D motion retrieval simply treat human motion as a uniform signal and do not consider the variations of motion between moments. Our core conceptual contribution is representing motion semantics as a composition of hierarchical frequency components. Low-frequency components encode long-term movement trends (e.g., trajectory), while high-frequency components capture fine-grained, short, and abrupt actions (e.g., sudden acceleration of a limb).
>
> Besides, our disordered motion sequence prediction module reflects the concept that temporal order implies semantic logic. We propose sequence ordering as a self-supervised learning objective to explicitly model the causal dynamics inherent in human motion.
>
> In summary, the significant performance gains are not only due to architecture, but also come from this new conceptual framework that aligns multi-scale motion frequencies with textual semantics.
>
> ---
>
> References
>
> [1] Exploring Vision Transformers for 3D Human Motion-Language Models with Motion Patches, CVPR 2024
>
> [2] Babel: Bodies, action and behavior with English labels, CVPR 2021
>
> [3] Generating diverse and natural 3d human motions from text, CVPR 2022
>
> [4] Tmr: Text-to-motion retrieval using contrastive 3d human motion synthesis, ICCV 2023
>
> [5] Intergen: Diffusion-based multi-human motion generation under complex interactions, IJCV 2024

---

> ### Author Response · Authors · 2025-11-22
> **Response to Reviewer j2hX (Part 2 of 2)**
>
> > **Question 1 - Robustness and generalization ability of WaMo on larger and more diverse datasets**
>
> To demonstrate the effectiveness and robustness of our method on longer and noisier motion sequences, **we further validate our method on the Motion-X++ dataset [1]**. Compared to HumanML3D and KIT-ML, it is a larger dataset, containing longer motion sequences with longer and noisier text annotations (generated by LLMs) in more diverse scenarios.
>
> We report results of the previous SOTA method MoPa [2], our base model without proposed modules (i.e., wavelet transforms and Disordered Motion Sequence Prediction), and our full setup in the table below:
>
> || Text-to-Motion ||| Motion-to-Text |||
> |-|-|-|-|-|-|-|
> ||R@1|R@2|R@3|R@1|R@2|R@3|
> | MoPa| 15.65| 22.64| 27.59| 15.63| 23.08| 27.82|
> | Base Model| 19.34| 27.54|33.14| 20.58| 29.09| 34.66|
> | **Base Model + Ours (WaMo)** | **24.11**| **33.11** | **39.20** | **24.19**| **34.84** | **40.85** |
>
> In particular, our method significantly outperforms MoPa. This is due to that MoPa pools joints into five body parts (torso, arms, legs), leading to significant information loss in dense skeletons (i.e., Motion-X++ with 52 joints compared to HumanML3D with 22 joints). In contrast, our method applies wavelet decomposition to the trajectory of every individual joint, preserving the fine-grained motion details and leading to better retrieval accuracy.
>
> > **Question 2 - Would it be possible to interpret the learned wavelet filters visually?**
>
> We thank the reviewer for this insightful suggestion. Interpreting the learned filters provides valuable insights into how our method adapts to the motion domain.
>
> We have visualized the scaling function $\phi(t)$ (low-pass) and wavelet function $\psi(t)$ (high-pass) before and after training. Please refer to Section D.7 in the revised PDF.
>
> The wavelet filters are initialized with the db1 wavelet, which consists of rigid box and step functions. After training on human motion data, the filters adapt significantly.
>
> The learned scaling function exhibits a non-uniform, decaying structure. It allows the model to better capture the smooth, low-frequency dynamics inherent in motion trajectories, instead of a simple moving average in standard db1 wavelets.
>
> The learned wavelet function transforms into a waveform with more oscillations. It captures complex high-frequency components of motions (e.g., sudden limb accelerations), which standard db1 wavelets might over-smooth.
>
> The visual comparison indicates that our learnable filters successfully adapt to the specific spatial-temporal characteristics of 3D human motion.
>
> ---
>
> References
>
> [1] Motion-X++: A Large-Scale Multimodal 3D Whole-body Human Motion Dataset, arXiv 2025
>
> [2] Exploring Vision Transformers for 3D Human Motion-Language Models with Motion Patches, CVPR 2024

---

### Author Response · Authors · 2025-12-03
**General Response**

We thank the reviewers and ACs for their valuable time and constructive comments on the paper.

In this paper, we propose the first multi-frequency analysis framework for Text-Motion Retrieval. We sincerely appreciate the reviewers’ recognition of our work in terms of:

- **Theoretical Grounding & Technical Soundness** (Reviewers j2hX, 3yLM). As Reviewer 3yLM stated: "The idea is intuitive and well-grounded, the technical design is elegant, addressing a key weakness of prior models".
- **Comprehensive Experiments** (Reviewers j2hX, 3yLM). As Reviewer j2hX stated: "The experimental settings and discussion are trustworthy and complete".
- **SOTA Performance** (all reviewers). Both Reviewers YAar and 1jTc stated: "Achieves clear empirical gains on standard benchmarks".

---

Below, we summarise the revisions we have provided to address reviewers' concerns. **All discussions have been incorporated into the revised manuscript and uploaded.**

- **Robustness and Generalization Ability** (Reviewers j2hX, YAar, 1jTc). We further validate our method on the Motion-X++ dataset [1] and two additional tasks, and extend our method to existing models.
- **Additional Visualizations and Analysis** (Reviewers j2hX, 3yLM). We have incorporated additional visualizations of learnable wavelet filters, t-SNE visualizations, and qualitative examples to justify our motivation.
- **Computational Analysis** (Reviewers YAar, 1jTc). We provide detailed comparisons on computational overhead.

---

We thank the reviewers and ACs again for the expertise and time. We appreciate the opportunity to strengthen our submission.

---

Reference

[1] Motion-X++: A Large-Scale Multimodal 3D Whole-body Human Motion Dataset, arXiv 2025

---

### Meta-Review · Area_Chair_aLXh · 2026-01-02

**Summary:**

This paper proposes a wavelet-based multi-frequency motion representation for fine-grained text-motion retrieval and reports strong empirical gains. Two reviewers were clearly positive (8) with relatively low confidence, emphasizing the technical soundness, and comprehensive evaluation.

The main concerns driving the mixed scores were (i) **conceptual novelty / retrieval-specific motivation** (whether wavelet decomposition + DMSP constitutes a substantially new contribution for TMR), (ii) baseline fairness / clarity, especially how MoMask is adapted to retrieval and why some reproduced results (e.g., LaMP) differ from prior reports, (iii) missing practical analysis such as compute/memory overhead, (iv) generalization beyond HumanML3D/KIT-ML, and (v) sensitivity/justification of Haar (db1) given the method’s “frequency” motivation.

**Reviewer Concerns:**

1. Conceptual novelty / retrieval-specific motivation (Reviewer YAar, 1jTc)
* **Outstanding**
* Authors provided task-specific framing and supporting ablations (e.g., wavelets vs DCT /FFT; DMSP vs video-style shuffling; and discussion of why time-localized semantics matter) However, the reviewers remain unconvinced as the frequency-based motion representation has been used by many other works.

2. Generalization beyond HumanML3D / KIT-ML; robustness on larger/noisier data and other tasks (Reviewer j2hX, YAar, 1jTc)
* **Addressed**
* Authors added Motion-X++ retrieval results and agued benefits on dense skeletons. They also demonstrated transfer to additional tasks (e.g., zero-shot motion classification/ interaction recognition) in response to generalization questions.

3. Baseline fairness/clarity: MoMask adaptation to retrieval, unexpected low LaMP results (Reviewer YAar, 1jTc)
* **Largely addressed**
* Authors explained the LaMP discrepancy as largely due to “Small Batches” vs standard retrieval protocol and provided supporting comparisons. Authors later provided training details (contrastive alignment, epochs/batch size/optimizer), and tested CLIP vs DistilBERT for MoMask to remove encoder inconsistency.

4. Computational overhead / scalability (params, memory, speed) (Reviewer YAar, 1jTc)
* **Addressed**
* Authors added a table reporting parameters, train/inference memory, training/inference time, and contrasted against MoPa/base.

5. Sensitivity to wavelet basis; why Haar (db1) is best (Reviewer YAar, 1jTc)
* **Partially addressed; still outstanding for YAar**
* Authors argued time-localization is more important for TMR semantics, provided cross-dataset consistency across wavelet choices, and discussed trade-offs. YAar’s follow-up challenged Haar’s poor frequency resolution and questioned whether it truly matches the paper’s frequency-domain motivation.

6.  Clarifications + additional visualizations (filters, Figure 1 interpretability, t-SNE, reconstruction quality, J definition) (Reviewer j2hX, 3yLM, 1jTc)
* **Addressed**
* Authors visualized learned wavelet filters (requested by j2hX) and expanded analysis/visuals.

**Reviewer Scores:**

Reviewer j2hX (rating 8 -> likely stay 8): their concerns are primarily about the score/impact rather soundness. The rebuttal adds extra-task evidence and filter visualizations. Reviewer j2hX rated with relatively low confidence, and quite short comments without ground evidence support.

Reviewer 3yLM (Rating 8 -> likely stay 8): the visible discussion suggests they asked for addition comparisons and visualizations which have been added in the rebuttal. Reviewer rated with relatively low confidence.

Reviewer YAar and 1jTc (Rating 4 0> likely stays 4): these two reviewers (with high confidence) share many concerns including novelty, unclear motivations, diversity. Though many of the questions (dataset diversity, momask applicability) were resolved, the key concerns regarding technical novelty and design remains. Therefore, it's not likely that they would flip the score.

Overall, given outweighed negative rating, I do not recommend acceptance of this submission.

---

### Decision · Program_Chairs · 2026-01-26

Reject